# Toward Physics-guided Time Series Embedding

## Abstract

In various scientific and engineering fields, the primary research areas have revolved around physics-based dynamical systems modeling and data-driven time series analysis. According to the embedding theory, dynamical systems and time series can be mutually transformed using observation functions and physical reconstruction techniques. Based on this, we propose Embedding Duality Theory, where the parameterized embedding layer essentially provides a linear estimation of the non-linear time series dynamics. This theory enables us to bypass the parameterized embedding layer and directly employ physical reconstruction techniques to acquire a data embedding representation. Utilizing physical priors results in a $10\times$ reduction in parameters, a $3\times$ increase in speed, and maximum performance enhancements of 18% in expert, 22% in zero-shot, and 53% in few-shot tasks without any hyper-parameter tuning. All methods are encapsulated as a plug-and-play module at `https://anonymous.4open.science/r/PSR-001/`.

## 1 Introduction

The explosion of real-time sensing data from the physical world opens up new opportunities for data-driven time series analysis, achieving widespread recognition in energy, transportation, education, meteorology, and other domains by leveraging the strong fitting capabilities of neural networks (Jin et al., 2024; Nie et al., 2022; Hu et al., 2024b; Mao et al., 2024). However, deep time series models struggle to comprehend the underlying physical laws of data, leading to a propensity for ***overfitting*** and ***lacking generalizability*** to unseen data (Zeng et al., 2023; Zhang et al., 2023; Hu et al., 2024a).

In numerous scientific and engineering disciplines, another central focus lies in dynamical systems that evolve over space and time, exampled in fluid mechanics, thermodynamics, and neuroscience (Brunton et al., 2020; Tan et al., 2023; Chen et al., 2021). According to the Takens (Takens, 1980) and Whitney theorems (Whitney, 1936), time series can be viewed as observations stemming from underlying dynamical systems, leading to a principal way to model the essence of time series data.

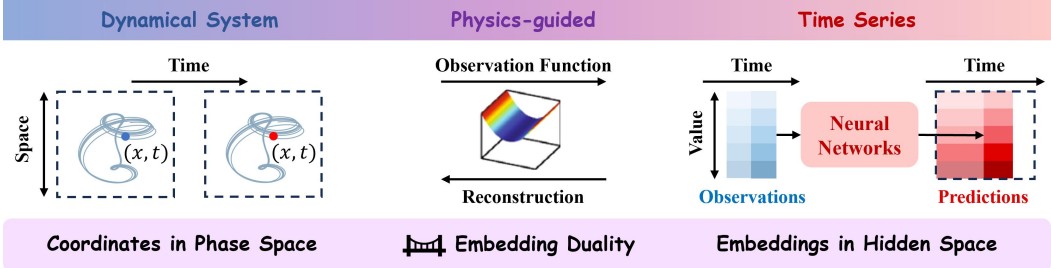

Figure 1: Dynamical systems embody physical laws unfolding in space and time, with time series as the low-dimensional observations. Our embedding duality theory bridges these two frameworks, demonstrating that parameterized hidden state representations are the model's estimation of dynamical system structures.

Our main goal is to develop a rich body of empirical and theoretical connections between the two frameworks. As illustrated in Figure 1, the dynamical system is primarily built on spatial coordinates sampled from physical equations, encapsulating the first-principle physical laws as they evolve over time. On the other hand, data-driven time series analysis first projects time series data into a high-dimensional latent space by a trainable embedding layer, relying on neural networks to model temporal dependencies based on the hidden representations. According to the Embedding Theory (Sauer et al., 1991), dynamical systems and time series can be mutually transformed through observation functions and numerical reconstruction techniques. Building on this inspiration, we introduce the concept of

*Embedding Duality*: *hidden state representation in deep time series model is equivalent to the underlying dynamical system structure of the data in phase space*. Theoretically, we demonstrate that parameterized embeddings serve as a linear estimation of underlying nonlinear dynamics, inheriting various physical properties of the system. Moreover, the feature space of system dynamics will transform into an ellipsoid space with model gradients. Empirically, Various experimental results, including dim scaling law, causal modeling, and visualizations, further support our propositions.

Supported by the Embedding Duality, we can skip parameterized embedding layers and directly apply physical priors and numerical techniques to reconstruct the underlying dynamical system structure as data embeddings (referred to as physics-guided time series embedding). Harnessing the powerful fitting capability of neural networks, we aim to subsequently parameterize the function-to-function dynamical system evolution within the Sobolev space for various downstream tasks. Leveraging the physical priors, as shown in Figure 2, where the octagon values quantify the performance improvement achieved through physics-guided embeddings, we have improved multiple model architectures on real-world time series analysis benchmarks. As immediate consequences of this paper:

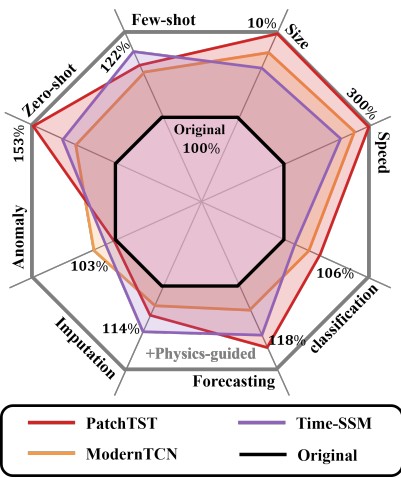

Figure 2: Performance comparison of physics-guided time series embedding versus original method across eight aspects.

- We innovatively integrate dynamical system Embedding Theory into the time series analysis tasks, bridging physical embedding and parameterized embedding from both theoretical and empirical perspectives. The *Embedding Duality* and various dynamical evidence are proposed.
- For expert models, which train from scratch on a specific dataset, we evaluate over ten embedding techniques, with our proposed physics-guided embedding achieving up to a **10×** reduction in parameters, a **3×** speed increase, an **18**% performance boost, and improved robustness across four time series analysis tasks and three neural network architectures. Notably, the physics-guided model reaches optimal performance **without** hyper-parameter tuning, while the carefully designed architecture yields marginal gains depending on specific hyper-parameters (Qiu et al., 2024).
- For foundation models, which leverage pre-training on diverse datasets or few samples, our methods lead to maximum improvements of **53**% in zero-shot and **22**% in few-shot tasks. This generality is expected to promote the emergence of physics-guided large-scale time series foundation models.

## 2 FORMULATION

**Definition 1** (**Dynamical System**). *Let the domain $S$ be an open subset of $\mathbb{R}^d$ and set an integer $k \geq 1$. Define the system state as $\boldsymbol{x} : S \mapsto \mathbb{R}^m$ where $\boldsymbol{x} = \left( x^1, \ldots, x^m \right)$. Then, an expression of:*

$$\mathcal{F} \left( D^k \boldsymbol{x}(s), D^{k-1} \boldsymbol{x}(s), \ldots, D\boldsymbol{x}(s), \boldsymbol{x}(s), s \right) = 0$$

*is called a $k^{th}$ order system of partial differential equation (or ordinary differential equation when $d = 1$), where $\mathcal{F} : \mathbb{R}^{md^k} \times \mathbb{R}^{md^{k-1}} \times \ldots \times \mathbb{R}^{md} \times \mathbb{R}^m \times S \mapsto \mathbb{R}^m$ and $s \in S$. Continuous systems typically exist on locally differentiable manifold spaces $\mathcal{M}$ Vlachos et al. (2008); Hu et al. (2024a).*

**Problem Statement.** Given multivariant historical sampled data $\boldsymbol{U} \in \mathbb{R}^{C \times T}$, the time series analysis model aims to derive a nonlinear functional mapping $f : \boldsymbol{U} \to \boldsymbol{Y}$ for various downstream tasks, e.g., forecasting, classification, anomaly detection, imputation. Adhere to the standard deep learning paradigm, $f$ can be decomposed into Embedding, Encoder, and Decoder parts, while in this paper:

(1) **Embedding** employs mathematical methods to reconstruct the underlying dynamical system based on time series data $\boldsymbol{U}$, 🔍 which is the research focus of this paper presented in Section 4.

(2) **Encoder** serves as a flexible architecture, with CNN-based (Luo & Wang, 2024), Transformer-based (Wen et al., 2023), and SSM-based (Hu et al., 2024b) models selected for this paper. Linear models (Zeng et al., 2023), which generally do not require embedding layers, fall outside the scope.

(3) **Decoder** follows mainstream time series model paradigms, utilizing token flattening and projection (Wang et al., 2024) operations to generate various output results depending on the task.

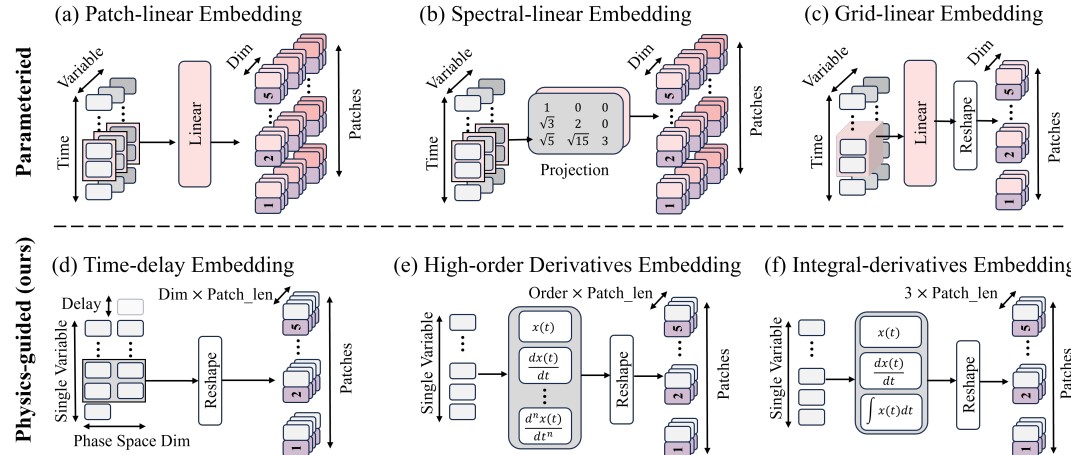

Figure 3: Existing parameterized embedding (a-c) and physics-guided non-parametric embedding (d-f) techniques. (a) Each time series patch utilizes a shared linear projection layer to obtain hidden representations. (b) Dense time series are processed using windowed spectral transformations with gradients for adaptability. (c) Multivariant time series are embedded using a shared linear layer on a grid measure. (d) Time Delay embedding based on predetermined hyper-parameters. (e) Higher-order derivative values are concatenated to reconstruct dynamical structures. (f) Integral terms can replace higher-order derivatives to address numerical instability.

## 3 RELATED WORK

**Parameterized Embedding in Deep Time Series Models.** The embedding technique, serving as a space transformation $\mathbb{R}^n \mapsto \mathbb{R}^m$, facilitates the mapping of discrete sparse features into continuous dense vector representations, laying a solid foundation for success across various machine learning domains. In time series analysis benchmark, mainstream methods utilize patch operation (Figure 3a) to conduct local linear projection (Nie et al., 2022), with certain models using convolutions to address inter-block information isolation (Hu et al., 2024c; Zhang et al., 2024b; Lin et al., 2024). Additionally, for audio modeling tasks, windowed spectral transformation techniques (Figure 3b), such as short-time Fourier transform, are commonly employed to extract time-frequency representations, thereby addressing concerns related to information density (Erol et al., 2024). Grid embeddings (Figure 3c) are commonly employed to handle the spatial-temporal relationships within the data (Gupta et al., 2021; Wu et al., 2023). Furthermore, several studies leveraged self-supervised learning to obtain enhanced embedding representations (Lee et al., 2023; Fraikin et al., 2023; Zhang et al., 2024a).

**Embedding Theory for Dynamics System Reconstruction.** Since significant results proposed by (Whitney, 1936; Takens, 1980) and formalized in (Sauer et al., 1991), the Embedding Theory has pervaded through almost all aspects of nonlinear dynamical systems (Definition 1). The time series $x(t) \in \mathbb{R}^n$ can be broadly interpreted as successive, though not always regular, observations of a dynamical system $\mathcal{F} \in \mathbb{R}^m$ via a measurement function $h : \mathbb{R}^m \mapsto \mathbb{R}^n (n < m)$. The main goal is to reconstruct the underlying system and explore its properties, which paved the way for developing numerous techniques like derivatives (Packard et al., 1980), integrals (Gilmore, 1998), time delay (Takens, 1980; Abarbanel et al., 1994), and principal component embedding (Broomhead & King, 1986). The system dynamics is subsequently learned using preferred modeling tools such as recurrent neural networks (Sangiorgio & Dercole, 2020), state space models (Alonso et al., 2024; Hu et al., 2024a), and reservoir computing (Haluszczynski & Räth, 2019; Yan et al., 2024), etc. Attraos (Hu et al., 2024a) pioneered using the time delay embedding technique in time series forecasting tasks, while our paper extensively explores various embedding techniques (Section 4.1), and provides comprehensive theoretical (Section 4.2) and experimental (Section 5) analysis.

## 4 PHYSICS-GUIDED TIME SERIES EMBEDDING

In this section, we commence by elucidating the implementation of our physic-guided embeddings, including the Time Delay, Principal Component, High-order Derivatives, and Integral-Differential methods. We summarize their properties before progressing to the proposed Embedding Duality.

### 4.1 TECHNICAL DETAILS

**Time Delay Embedding (TD-Emb).** As shown in Figure 1(d), time delay embedding augments a scalar time series $x \in \mathbb{R}^T$ into a higher-dimensional dynamical system $\mathcal{F} \in \mathbb{R}^{m \times (T-(m-1)\tau)}$, where $\mathcal{F}(t) = (x(t), x(t-\tau), \ldots, x(t-(m-1)\tau))$, by embedding dimension $m$ and time delay

Table 1: Comprehension comparison for various physics-guided embedding methods.

| Method | Interpretability | Performance | Robustness | Convergence Rate | Hyper-parameter |
|---|---|---|---|---|---|
| **🔍 Time Delay (TD-Emb)** | **Best** | Good | Good | **Best** | $m, \tau \leftarrow$ physical prior |
| **🔍 High-order Derivatives (HD-Emb)** | Good | **Best** | **Best** | Good | $m, \Delta \leftarrow$ typically (3,1) |
| Integral-differential (ID-Emb) | Trivial | Trivial | Good | Good | $\Delta \leftarrow$ typically (1) |
| Principal Component (PC-Emb) | Trivial | Poor | Poor | Poor | $m, k \leftarrow$ physical prior |

$\tau$. Theoretically, when $m$ exceeds twice the dynamical dimension, the homeomorphic structure can be reconstructed (Vlachos et al., 2008). In this paper, $\tau$ is set heuristically as a quarter of the most dominant period in the signal[1] and $m$ is determined by the CC method (Kim et al., 1999).

For multivariant time series, guided by the Lyapunov exponents of each variable, we can either employ a channel-independent strategy (CI) or concatenate $\{\mathcal{F}_i\}_{i=1}^C$ (Vlachos et al., 2008) as a whole to employ a channel-dependent strategy (CD). This will be omitted in the following descriptions.

**Principal Component Embedding (PC-Emb).** TD-Emb is the most popular method for visualizing the dynamical structures of systems within Euclidean space. However, its performance is highly sensitive to the choice of hyper-parameters. As an alternative, PC-Emb, outlined in Eq. 1, begins by applying TD-Emb to obtain $X$, followed by the computation of the covariance matrix $C$ from $X$. Finally, a $k$-dimensional Principal Component Analysis (PCA) is performed to derive the system representation $\mathcal{F}$. Where $X \in \mathbb{R}^{m \times (T-(m-1))}$, $C \in \mathbb{R}^{m \times m}$, and $\mathcal{F} \in \mathbb{R}^{m \times k}$.

$$X = \text{TD-Emb}(m, \tau = 1, x) \qquad C_{ij} = \langle X_{ij} \rangle \qquad \mathcal{F} = \text{PCA}(k, C) \tag{1}$$

**High-order Derivatives Embedding (HD-Emb).** In addition to the TD-Emb method, we leverage the multi-order characteristics of the system in Definition 1 by directly concatenating high-order derivatives to construct $\mathcal{F} \in \mathbb{R}^{(m+1) \times T}$. In Eq. 2, we utilize the Forward Differencing technique to approximate this continuous process, with hyper-parameters: order $m$ and discrete step size $\Delta$.

$$\mathcal{F}(t) = \left( x(t), \frac{dx(t)}{dt}, \ldots, \frac{d^m x(t)}{dt^m} \right) \textbf{ (Continuous)} \qquad \frac{dx(t)}{dt} \approx \frac{x(t+\Delta) - x(t)}{\Delta} \textbf{ (Discrete)} \tag{2}$$

Although $m$ and $\Delta$ can still be calculated using numerical methods (Tan et al., 2023), our experiment results indicate that $m = 3$ and $\Delta = 1$ generally yield the best performance. In some studies related to ordinary differential equations and state-space models (Smith et al., 2022; Gu & Dao, 2023; Hu et al., 2024b), $\Delta$ is defined as a learnable parameter to selectively emphasize important information in the data. In this research, we prioritize the efficiency of non-parameterized physical priors, while the exploration of trainable High-order Derivatives embedding is left for future work.

**Integral-differential Embedding (ID-Emb).** However, in the HD-Emb method, the approximations of successive higher-order derivatives are generally negatively impacted by the signal-to-noise ratio. In Eq. 3, an alternative method is to replace the high-order terms with the integral value by only hyper-parameter $\Delta$, where continuous integration can be approximated using summation operations.

$$\mathcal{F}(t) = \left( \int_{-\infty}^{t} x(t) dt, x(t), \frac{dx(t)}{dt} \right) \textbf{ (Continuous)} \qquad \int_{-\infty}^{t} x(t) dt \approx \Delta \sum_{i=1}^{T} x(t + i\Delta) \textbf{ (Discrete)} \tag{3}$$

**Patch & Padding.** In order to reduce computational complexity and enhance model stability, we adhere to mainstream practices (Nie et al., 2022) by segmenting the obtained dynamical system $\mathcal{F}$ using two parameters: patch length and stride. For lengths that are not divisible, we employ the left zero-padding operation, which achieves optimal performance compared to other padding types.

**Discussion.** As shown in Table 1, we provide a comprehensive comparison of the four methods. Both TD-Emb and HD-Emb achieve optimal performance in two metrics each. In contrast, ID-Emb is constrained by a fixed dimension of 3, and PC-Emb tends to lose critical nonlinear information during the PCA process, resulting in poor performance. Consequently, the experimental section will primarily focus on the first two methods. Furthermore, we have encapsulated various embedding methodologies within the code repository, enabling direct invocation with ***a single line of code***.

### 4.2 THEORETICAL JUSTIFICATION FOR EMBEDDING DUALITY

**Proposition 4.1.** *The embedding method, which uses a shared linear matrix, is an integral transformation $h(t) = \int_{-\infty}^{t} x(s)\phi(t, s) d\mu(s)$ with limited time-invariant measure $\mu$ and polynomial basis $\phi$.*

---

[1]The sine wave $x(t) = \sin(\omega t)$ yields the most circular embedding in a 2D plane with $\tau = 2\pi/4\omega$

(a) Parameterized Embedding as Dynamics Estimator

(b) Directly applying physical priors as dynamics.

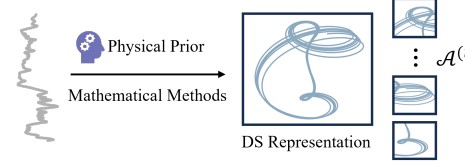

Figure 4: Illustration for (a) Parameterized embedding and (b) Physics-guided embedding technology.

Considering polynomials as universal approximators for dynamical systems (Bollt, 2021), Proposition 4.1 indicates that parameterized embedding methods (Figure 3(a-c)) essentially projects input time series into polynomial spectral space to represent the dynamical structure, where the measure $\mu$, or weight function sometimes, is governed by patch length, $\phi$ is parameterized by the embedding matrix.

**Proposition 4.2.** *For the full-rank embedding matrix, the embedding process is a similarity transformation that maintains the original dynamical properties (system eigenvalues).*

Typically, the dense matrix in deep models is considered to be full rank. Proposition 4.2 shows that the parameterized embedding process is a mere space coordinate transformation, with non-linear time series dynamics linearized by the average Jacobin value within the data patch.

**Lemma 4.3.** *A continuous function $f$ is $K$-Lipschitz when $\|f(x_1) - f(x_2)\| \leq K\|x_1 - x_2\|$, then:*

*(1) The state space model with the negative diagonal matrix $\mathbf{A}$ and normalization layers is 1-Lipschitz.*

*(2) The fully connected and convolution neural network with normalization layers is 1-Lipschitz.*

*(3) The standard dot-product attention is not Lipschitz. The $L_2$ attention is bounded Lipschitz.*

The Lipschitz continuity restricts the dynamical structure under small perturbations, ensuring that when $K$=1, the dynamical properties are generally preserved. Lemma 4.3 allows us to disregard the influence of the encoder architecture in most cases, even though the transformer backbone may not always be optimal, to focus exclusively on the dynamical changes within the embedding layer. Moreover, it provides a solid foundation for flexibly replacing the embedding layer as needed.

**Dynamical Feature Space** As illustrated in Figure 4(a), for the embedding projection matrix $V = \text{eig}(v^1, \cdots, v^M)$ initialized with a normal distribution, when the dimension is sufficiently large, its feature space can be considered spherical. For the gradient $\nabla$ passed into the embedding layer, we can apply the singular value decomposition (SVD) with the diagonal matrix $S = \text{diag}(\sigma_1, \cdots, \sigma_M)$ and orthogonal matrix $U = (u^1, \cdots, u^M)$, specifically, $\nabla v^m = \sigma u^m$. This process can be described as the transformation of a spherical feature space (slice) into an ellipsoidal feature space.

**Lemma 4.4.** *The Lyapunov exponents $\lambda_m = \lim_{t \to \infty} \frac{1}{t} \ln \sigma_m(t)$ of the system attractors are the mean logarithmic growth rates of the principal axes lengths of the ellipsoidal feature space.*

According to Lemma 4.4, the attractors of the system, which represent underlying data patterns, are reflected in the lengths of the axes within the ellipsoidal feature space. In Figure 4(a), scaling the embedding layer's feature space using model gradients can be interpreted as an adaptive estimation of the underlying dynamical structure, where the system attractor is captured through the logarithms of the eigenvalues of the Oseledec matrix (ose, 1968). In contrast, as demonstrated in Figure 4(b), our proposed physics-guided embeddings bypass this adaptive scaling process. Rather than relying on gradient-based adjustments, they reconstruct the system's dynamical trajectory using physical priors and numerical methods, obtaining the attractor representation directly through patch operations. This provides a more efficient and interpretable means to capture the system's dynamics.

**Dynamical System Characteristics** According to the Embedding Theory, insufficient phase space dimensions cause dynamical structures to stack, obscuring their true shapes. Conversely, excessive dimensions expand the structure excessively, amplifying noise effects. The threshold typically equals twice the latent dynamics, and exceeding this threshold results in spurious structures. Based on this, we propose the following conjectures, which are empirically validated in Section 5.1.

**Conjecture 4.5** (**Dim Scaling Law**). *For parameterized embedding, as the hidden dimensions increase, the model loss will generally decrease initially and then increase, as shown in Figure 6.*

**Conjecture 4.6** (**Spurious Dynamics**). *Bidirectional modeling, whether through a transformer or SSM backbone, helps eliminate spurious dynamical structures that are sensitive to the positional inductive bias, consequently enhancing performance as empirically demonstrated in Table 2.*

## 5 EXPERIMENTS

In this section, we focus on addressing the following research questions: ***RQ1***: Does the Embedding Duality theory empirically exist? ***RQ2***: How do physics-guided embeddings perform in expert tasks, how do they achieve this, and do they have the potential to become a new embedding paradigm? ***RQ3***: How do physics-guided embeddings perform in foundation tasks, and do they have the potential to be transferred to large-scale time series foundational models? All Experiments are based on the Time Series Library (Wang et al., 2024), details regarding model backbones, experimental settings, full results, technical backgrounds, and further inspirations are reported in Appendix C.

### 5.1 EMPIRICAL EVIDENCE FOR EMBEDDING DUALITY

**Visualization.** As illustrated in Figure 5, we present the visual representations of previous parameterized embeddings (left) and our proposed physics-guided embeddings (right: TD-Emb) when utilizing PatchTST as the model backbone. It is evident that the patterns exhibited by these two types of embeddings bear a remarkable resemblance. For instance, in Figure 5(a), the periodic patterns captured by the parameterized embeddings align with the consistent stripes present in dynamical structures, whereas in Figure 5(b), the smooth regions depicted in the parameterized embeddings correspond to the band-like dark regions in the dynamical structure. Notably, ***the data patterns captured by the physics-guided embedding have been significantly enhanced, highlighting their superior performance in various downstream tasks.***

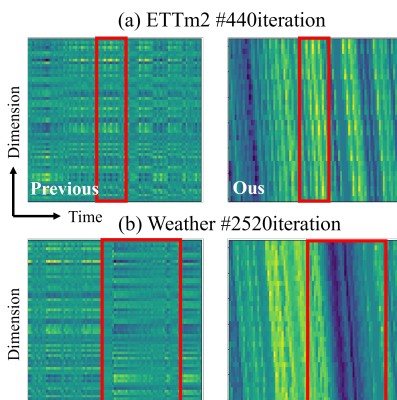

Figure 5: Embedding Visualizations.

**Dim Scaling Law.** In Figure 6, we present the correlation between average model performance and hidden layer dimensions based on PatchTST backbone across three datasets (ETTm2, ETTh2, Weather). Consistent with the proposition 4.5, we observe a decrease followed by an increase in the MSE loss. Moreover, the optimal performance occurs within a specific range, typically 1-2 times the underlying dynamical dimension associated with the dataset. For example, considering a physical prior dimension of 4 for the ETTm2 dataset, where the patch length is 16, yielding a total dimension of 64, the optimal interval lies in 64-128. This remarkable discovery shows that ***parameterized embeddings have effectively encapsulated the intrinsic dynamical characteristics of the data.***

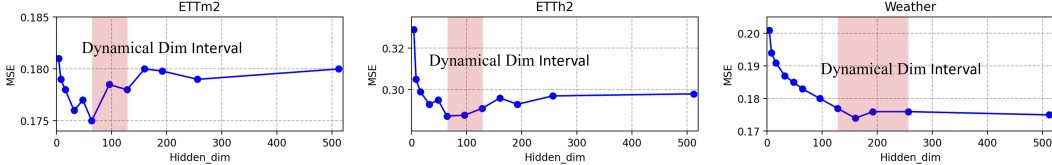

Figure 6: Dim scaling laws: The shaded region represents the ideal dimension interval of the system dynamics.

**Causal-directional Modeling.** In Table 2, we explore the effects of unidirectional and bidirectional modeling, termed causal-directional modeling, on the performance of the Time-SSM and PatchTST models. Specifically, we adapt the attention mechanism with a causal mask and combine the SSM outcomes bidirectionally using a linear layer, respectively. Our results reveal the following insights: (a) Bidirectional modeling typically outperforms unidirectional modeling by a consistent margin, as supported by Proposition 4.6. (b) The performance gap is more pronounced in the PatchTST model, possibly attributed to SSM's adeptness in capturing dynamical structures, hence mitigating the impact of incidental dynamics. (c) Model variations incorporating physics-guided embeddings contribute to mitigating performance differentials between unidirectional and bidirectional modeling approaches.

Table 2: Average performance comparison for causal-directional modeling validation. The improved results are highlighted in ▨. M1: Time-SSM; M2: PatchTST; -U: Unidirectional; -B: Bidirectional.

| | M1-U | | **M1-B** | | M1-TD-U | | **M1-TD-B** | | M1-HD-U | | **M1-HD-B** | | M2-U | | **M2-B** | | M2-TD-U | | **M2-TD-B** | | M2-HD-U | | **M2-HD-B** | |
|---|---|---|---|---|---|---|---|---|---|---|---|---|---|---|---|---|---|---|---|---|---|---|---|---|
| | MSE | MAE | MSE | MAE | MSE | MAE | MSE | MAE | MSE | MAE | MSE | MAE | MSE | MAE | MSE | MAE | MSE | MAE | MSE | MAE | MSE | MAE | MSE | MAE |
| ETTh1 | 0.439 | 0.438 | 0.436 | 0.437 | 0.436 | 0.436 | 0.434 | 0.433 | 0.432 | 0.435 | 0.435 | 0.437 | 0.456 | 0.453 | 0.450 | 0.449 | 0.446 | 0.448 | 0.438 | 0.440 | 0.447 | 0.450 | 0.437 | 0.437 |
| ETTh2 | 0.379 | 0.401 | 0.371 | 0.394 | 0.376 | 0.399 | 0.371 | 0.393 | 0.374 | 0.398 | 0.369 | 0.391 | 0.389 | 0.416 | 0.382 | 0.411 | 0.383 | 0.407 | 0.376 | 0.401 | 0.381 | 0.404 | 0.374 | 0.398 |
| ETTm1 | 0.389 | 0.403 | 0.388 | 0.404 | 0.387 | 0.402 | 0.388 | 0.404 | 0.384 | 0.400 | 0.386 | 0.402 | 0.392 | 0.405 | 0.388 | 0.402 | 0.387 | 0.399 | 0.385 | 0.396 | 0.379 | 0.392 | 0.378 | 0.393 |
| ETTm2 | 0.284 | 0.330 | 0.281 | 0.297 | 0.285 | 0.330 | 0.282 | 0.329 | 0.282 | 0.328 | 0.285 | 0.331 | 0.287 | 0.331 | 0.291 | 0.334 | 0.279 | 0.325 | 0.281 | 0.328 | 0.285 | 0.331 | 0.283 | 0.330 |

## 5.2 Performance for Expert Models.

**Forecasting.** We maintain the model hyper-parameters (detailed in Appendix C.3) to conduct a fair comparison for previous parameterized and our proposed physics-guided embeddings across diverse model architectures. As depicted in Table 3, the following observations can be made: (a) Generally, the incorporation of physical priors significantly boosts forecasting performance, with the most substantial gain being 11% on the Exchange dataset. (b) The HD-Emb typically delivers top performance and, due to its efficiency, is expected to become the standard embedding technology for expert models. (c) The PatchTST model shows the most significant performance improvement among the three architectures, followed by Time-SSM and ModernTCN. This could be attributed to Theorem 4.3, suggesting that the standard dot-product attention lacks Lipschitz continuity and struggles to adaptively capture the underlying dynamics, while the physical priors effectively resolve this issue. (d) *Recently, community efforts have primarily focused on developing more advanced encoder architectures; however, the improvements achieved are minimal (Qiu et al., 2024). In contrast, our proposed plug-and-play module demonstrates a significant enhancement in performance.*

Table 3: Average Performance for long-term forecasting task with input length is 96 and CI modeling strategy. The first and second results are highlighted in ▢ and ▢. Full results are reported in Table 13.

| | Time-SSM | | +TD | | +HD | | PatchTST | | +TD | | +HD | | ModernTCN | | +TD | | +HD | |
|---|---|---|---|---|---|---|---|---|---|---|---|---|---|---|---|---|---|---|
| | MSE | MAE | MSE | MAE | MSE | MAE | MSE | MAE | MSE | MAE | MSE | MAE | MSE | MAE | MSE | MAE | MSE | MAE |
| ETTh1 | 0.439 | 0.438 | 0.436 | 0.436 | 0.432 | 0.435 | 0.450 | 0.449 | 0.438 | 0.440 | 0.437 | 0.437 | 0.445 | 0.432 | 0.441 | 0.423 | 0.440 | 0.429 |
| ETTh2 | 0.379 | 0.401 | 0.375 | 0.399 | 0.374 | 0.398 | 0.382 | 0.411 | 0.376 | 0.401 | 0.374 | 0.398 | 0.382 | 0.404 | 0.378 | 0.400 | 0.376 | 0.400 |
| ETTm1 | 0.389 | 0.403 | 0.387 | 0.402 | 0.384 | 0.400 | 0.388 | 0.402 | 0.385 | 0.396 | 0.378 | 0.393 | 0.386 | 0.401 | 0.387 | 0.402 | 0.382 | 0.398 |
| ETTm2 | 0.284 | 0.330 | 0.285 | 0.330 | 0.282 | 0.328 | 0.291 | 0.334 | 0.281 | 0.328 | 0.283 | 0.330 | 0.285 | 0.327 | 0.290 | 0.330 | 0.286 | 0.326 |
| ECL | 0.203 | 0.289 | 0.203 | 0.288 | 0.201 | 0.289 | 0.204 | 0.294 | 0.214 | 0.308 | 0.204 | 0.299 | 0.215 | 0.293 | 0.217 | 0.297 | 0.213 | 0.291 |
| Exchange | 0.367 | 0.407 | 0.357 | 0.402 | 0.355 | 0.400 | 0.393 | 0.419 | 0.358 | 0.403 | 0.361 | 0.404 | 0.393 | 0.425 | 0.373 | 0.412 | 0.377 | 0.415 |
| Weather | 0.254 | 0.279 | 0.251 | 0.276 | 0.254 | 0.278 | 0.258 | 0.280 | 0.253 | 0.278 | 0.258 | 0.282 | 0.243 | 0.273 | 0.242 | 0.275 | 0.238 | 0.268 |

**Forecasting *w.r.t.* Efficiency.** As depicted in Figure 7, we present an efficiency visualization of various model architectures in the ETTh1 dataset. Key observations include: (a) Physics-guided embeddings bypass embedding matrices and reduce model dimensions, leading to a significant reduction in parameter count across various architectures (e.g., the PatchTST model exhibits a 10× reduction), alongside performance improvements. (b) The necessity of deep neural networks in time series analysis tasks has long been debated, as some linear models have achieved strong results with greater efficiency (Zeng et al., 2023; Xu et al., 2023;

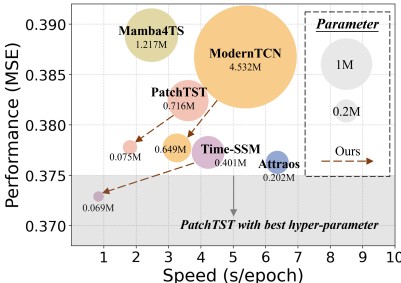

Figure 7: Efficiency analysis in ETTh1.

Lin et al., 2024). Our proposed method directly aligns the parameter count of existing deep time series models to that of linear models with superior performance, which marks *physics-guided Embs, especially HD-Emb, to potentially become the standard embedding method for expert models*.

**Forecasting *w.r.t.* Input Length.** In accordance with Table 4, we investigate the impact of input length on performance. It is observed that: (a) Across various lengths, the physics-guided embeddings consistently enhance performance, with HD-Emb exhibiting the best performance. Moreover, as the input length increases, the enhancement becomes more pronounced. This phenomenon is attributed to the fact that in embedding theory, longer input time series can better reconstruct the underlying dynamical structure, with a length of 1000 typically considered sufficient for ideal structural reconstruction. (b) The limitation of the Transformer architecture in modeling *long-range dependencies* has long been challenged (Nie et al., 2022), as model performance tends to degrade with input length over 336. However, our physics-guided embeddings offer a solution to this issue.

Table 4: Average forecasting performance *w.r.t.* input lengths. The improved and decreased results are highlighted in ▢ and ▢; improvements exceeding 10% are highlighted in ▢. Full results are reported in Table 11.

| | Original-96 | | +TD | | +HD | | Original-336 | | +TD | | +HD | | Original-720 | | +TD | | +HD | |
|---|---|---|---|---|---|---|---|---|---|---|---|---|---|---|---|---|---|---|
| | MSE | MAE | MSE | MAE | MSE | MAE | MSE | MAE | MSE | MAE | MSE | MAE | MSE | MAE | MSE | MAE | MSE | MAE |
| ETTh1 | 0.450 | 0.449 | 0.438 | 0.440 | 0.437 | 0.437 | 0.420 | 0.441 | 0.417 | 0.432 | 0.413 | 0.422 | 0.481 | 0.483 | 0.428 | 0.449 | 0.495 | 0.482 |
| Improve | – | – | 2.67% | 2.00% | 2.89% | 2.67% | – | – | 0.71% | 2.04% | 1.67% | 4.31% | – | – | 11.02% | 7.04% | -2.91% | 0.21% |
| ETTh2 | 0.382 | 0.411 | 0.375 | 0.399 | 0.374 | 0.398 | 0.358 | 0.415 | 0.354 | 0.397 | 0.363 | 0.400 | 0.439 | 0.447 | 0.357 | 0.405 | 0.361 | 0.408 |
| Improve | – | – | 1.83% | 2.92% | 2.09% | 3.16% | – | – | 1.12% | 4.34% | -1.40% | 3.61% | – | – | 18.68% | 9.40% | 17.77% | 8.72% |
| ETTm2 | 0.284 | 0.330 | 0.285 | 0.330 | 0.282 | 0.328 | 0.263 | 0.323 | 0.262 | 0.321 | 0.259 | 0.319 | 0.280 | 0.339 | 0.275 | 0.337 | 0.273 | 0.331 |
| Improve | – | – | 0.35% | – | 0.70% | 0.61% | – | – | 0.38% | 0.62% | 1.52% | 1.24% | – | – | 1.79% | 0.59% | 2.50% | 2.36% |
| Weather | 0.254 | 0.279 | 0.251 | 0.276 | 0.254 | 0.278 | 0.233 | 0.270 | 0.231 | 0.267 | 0.230 | 0.265 | 0.231 | 0.273 | 0.227 | 0.267 | 0.222 | 0.260 |
| Improve | – | – | 1.18% | 1.08% | – | 0.36% | – | – | 0.86% | 1.11% | 1.29% | 1.85% | – | – | 1.73% | 2.20% | 3.90% | 4.76% |

**Forecasting *w.r.t.* Various Embedding Techniques.**   As shown in Figure 8, we evaluate the performance of 10 embedding methods across 7 datasets. The striped bars represent the CD modeling strategy, which combines dynamical structures from multiple variables into a unified system. Key observations include: (a) Physics-guided embeddings (blue) generally outperform parameterized embeddings (red), with the HD-Emb performing the best overall, followed by the TD-Emb and ID-Emb methods. (b) Significant improvements are observed in datasets with fewer variables, like ETT, and those with strong physical characteristics, like sunspots. (c) CD strategy yields substantial improvements in datasets such as ETT#2, Weather, and ECL, but has adverse effects in ETT#1. This is attributed to data characteristics; for instance, Weather variables show similar Lyapunov and mutual information indices, indicating a shared underlying dynamical system, unlike the diverse indices in ETT#1, which favors channel-independent modeling. (d) Grid embedding performs poorly, likely due to the need to concatenate multivariate data in dynamical space, hindering the capture of system dynamics in the temporal domain. (e) Spectral embedding is also suboptimal, as time-series data is less dense than audio and spectral transformations like STFT may disrupt temporal sequencing.

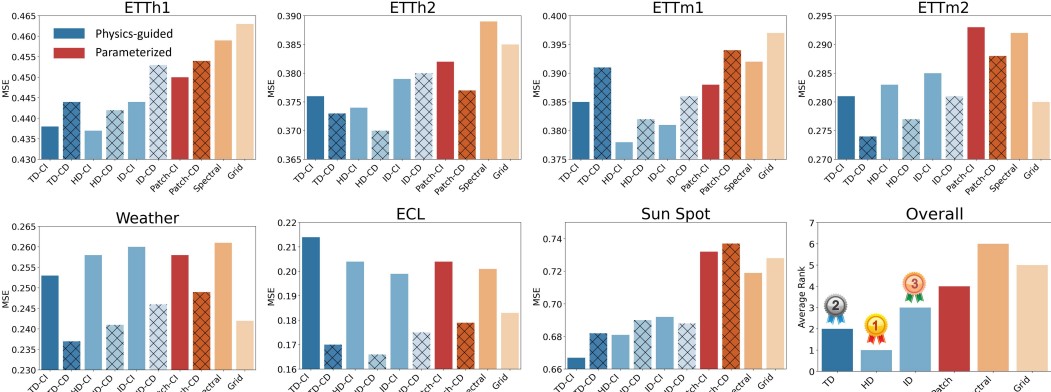

Figure 8: Average forecasting performance comparison for various embedding methods.

**Forecasting *w.r.t.* Testing Curve.**   As depicted in Figure 9, we depict the fluctuations in test loss for the original PatchTST model and its variant integrating physics-guided embeddings (TD-test and HD-test). Noteworthy observations include: (a) Generally, data representations originating from physics-guided embeddings exhibit more consistent gradients and attain superior fitting accuracy due to the physical prior. Conversely, parameterized embeddings often grapple with ***overfitting issues***, thereby illuminating the heightened efficacy of physics-guided embeddings.

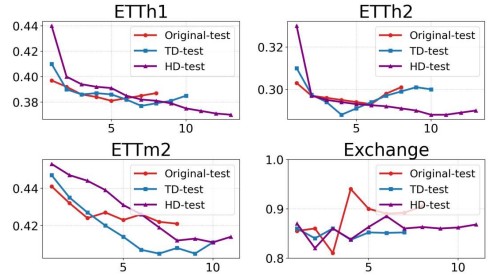

Figure 9: Visualization of testing loss & epochs.

(b) ***High-order derivatives embedding manifests the most stable gradients and the slowest convergence rate throughout the dataset, enabling a gradual advancement toward the optimal solution.***

**Forecasting *w.r.t.* Robustness Analysis.**   As illustrated in Figure 10, we conduct robustness analyses using five experimental hyper-parameters across four datasets and three input lengths. The key observations include: (a) Compared to parameterized embeddings, physics-guided methods have significantly improved robustness, with this advantage further amplifying as the input length increases. (b) The parameterized embedding struggles to leverage longer time series context. Conversely, physics-guided embeddings demonstrate a more consistently increasing performance with longer input length. (c) Parameterized embeddings exhibit significant variability. Although recent models assert state-of-the-art (SOTA) results,

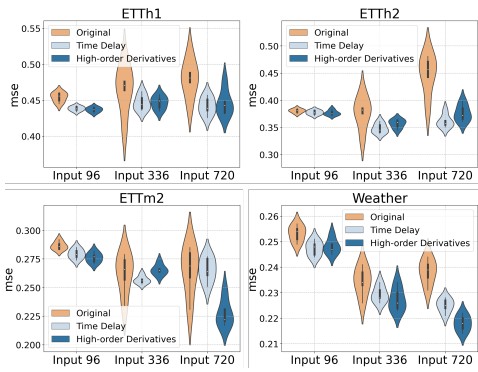

Figure 10: Robustness of PatchTST backbone.

they rely heavily on precise hyper-parameters, whereas our proposed ***physics-guided embeddings consistently maintain a good performance without meticulous hyper-parameter searching.***

**Classification.** As shown in Table 5, we conduct an investigation in the realm of classification tasks. Our findings reveal that: (a) Akin to discoveries in the field of neuroscience (Chen et al., 2021), dynamical structures play a significant facilitative role in classification tasks, leading to an enhancement in performance compared to parameterized embedding across various architectures, especially for the Time Delay embedding. b) The High-order Derivatives embedding is suboptimal, and we suspect this lackluster performance may stem from the inherent smoothing nature of derivative operations, potentially leading to the loss of information beneficial for classification tasks.

Table 5: Performance comparison for the classification task based on the hyper-parameters provided in the original paper. The first and second results are highlighted in ▮ and ▮; OOM means out of memory.

| Datasets / Models | Time-SSM | TD-Emb | HD-Emb | PatchTST | TD-Emb | HD-Emb | M-TCN | TD-Emb | HD-Emb |
|---|---|---|---|---|---|---|---|---|---|
| EthanolConcentration | 0.311 | 0.321 (3.22%) | 0.311(0.00%) | 0.307 | 0.324 (5.54%) | 0.311 (1.30%) | 0.319 | 0.333 (4.39%) | 0.312 |
| FaceDetection | 0.673 | 0.681 (1.19%) | 0.655 | 0.681 | 0.659 | 0.638 | 0.687 | 0.694 (1.02%) | 0.665 |
| Handwriting | 0.279 | 0.289 (3.58%) | 0.261 | 0.286 | 0.295 (3.15%) | 0.245 | 0.284 | 0.292 (2.82%) | 0.263 |
| Heartbeat | 0.714 | 0.737 (3.22%) | 0.702 | 0.736 | 0.749 (1.77%) | 0.707 | 0.771 | 0.778 (0.91%) | 0.727 |
| JapaneseVowels | 0.974 | 0.981 (0.72%) | 0.922 | 0.957 | 0.977 (2.09%) | 0.955 | 0.981 | 0.986 (0.51%) | 0.967 |
| PEMS-SF | OOM | OOM | OOM | 0.861 | 0.879 (2.09%) | 0.818 | 0.832 | 0.857 (3.00%) | 0.822 |
| SelfRegulationSCP1 | 0.870 | 0.893 (2.64%) | 0.872 (0.23%) | 0.896 | 0.903 (0.78%) | 0.913 (1.90%) | 0.928 | 0.934 (0.65%) | 0.905 |
| SelfRegulationSCP2 | 0.589 | 0.572 | 0.607 (3.06%) | 0.577 | 0.565 | 0.595 (3.12%) | 0.617 | 0.622 (0.81%) | 0.620 (0.49%) |
| SpokenArabicDigits | 0.980 | 0.994 (1.43%) | 0.983 (0.31%) | 0.959 | 0.983 (2.50%) | 0.978 (1.98%) | 0.981 | 0.979 | 0.966 |
| UWaveGestureLibrary | 0.834 | 0.853 (2.28%) | 0.805 | 0.853 | 0.838 | 0.859 | 0.859 (0.70%) | 0.866 (0.81%) | 0.844 |

**Imputation & Anomaly Detection.** As shown in Figure 11, we present the performance analysis of Imputation and Anomaly Detection tasks. Overall, physics-guided embeddings consistently improve performance on the Imputation task, with particularly notable enhancements for the SSM-based backbone (14.7% in ETTh2 dataset). However, for the Anomaly Detection task, the impact of physics-guided embeddings on performance is minimal, except for the SWAT dataset.

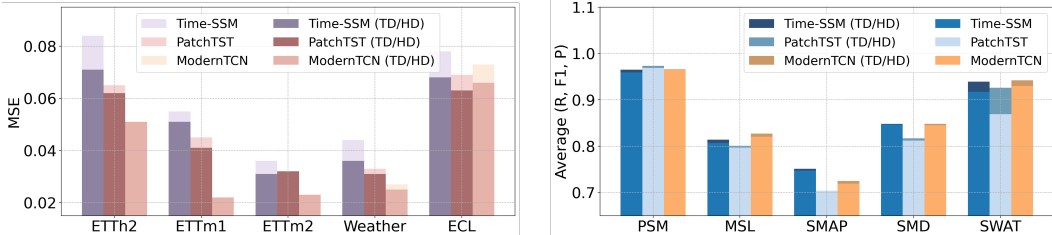

Figure 11: Average performance comparison for Imputation (left) and Anomaly Detection (right) tasks.

**Tasks Summary.** For information-intensive tasks such as forecasting and imputation, physics-guided embeddings can better comprehend the underlying dynamical characteristics and exhibit robustness, leading to significant performance improvements. For non-information-intensive tasks like classification, the dynamical structures constructed by TD-Emb methods can provide physics-related features that deep learning might overlook, thereby enhancing performance, while the HD-Emb method may lose some crucial information. In anomaly detection tasks, which may rely more on periodicity and data distribution, the impact of physics-guided embeddings is less pronounced.

## 5.3 PERFORMANCE FOR FOUNDATION MODELS

**Few-shot Learning.** As illustrated in Table 6, it can be observed that physics-guided embeddings yield stable and significant performance improvements, with a maximum performance boost of 21% on the Time-SSM architecture. Consistent with the forecasting task, the HD method demonstrates the best performance, closely followed by the TD method. We attribute this to the more pronounced physical characteristics of the dynamical system compared to temporal data features, as depicted in Figure 5. Therefore, in the few-shot learning, ***physics-guided embeddings have the capacity to encapsulate richer and more essential information, consequently amplifying the performance.***

Table 6: Average Few-shot results on 10% training data with input 336. The improved and decreased results are highlighted in ▮ and ▮; improvements over 10% are highlighted in ▮. Full results are reported in Table 12.

| | Time-SSM | | +TD | | +HD | | PatchTST | | +TD | | +HD | | ModernTCN | | +TD | | +HD | |
|---|---|---|---|---|---|---|---|---|---|---|---|---|---|---|---|---|---|---|
| | MSE | MAE | MSE | MAE | MSE | MAE | MSE | MAE | MSE | MAE | MSE | MAE | MSE | MAE | MSE | MAE | MSE | MAE |
| ETTh1 | 0.763 | 0.617 | 0.695 | 0.550 | 0.603 | 0.523 | 0.664 | 0.567 | 0.585 | 0.524 | 0.567 | 0.515 | 0.582 | 0.525 | 0.542 | 0.504 | 0.533 | 0.497 |
| Improve | – | – | 8.91% | 10.86% | 20.97% | 15.24% | – | – | 11.90% | 7.58% | 14.61% | 9.17% | – | – | 6.87% | 4.00% | 8.42% | 5.33% |
| ETTh2 | 0.515 | 0.485 | 0.498 | 0.479 | 0.496 | 0.481 | 0.449 | 0.448 | 0.412 | 0.431 | 0.398 | 0.426 | 0.390 | 0.414 | 0.397 | 0.413 | 0.384 | 0.412 |
| Improve | – | – | 3.30% | 0.12% | 3.69% | 0.82% | – | – | 8.24% | 3.79% | 11.36% | 4.91% | – | – | -1.79% | 0.24% | 1.54% | 0.48% |
| ETTm2 | 0.342 | 0.370 | 0.322 | 0.360 | 0.299 | 0.346 | 0.300 | 0.340 | 0.284 | 0.332 | 0.284 | 0.334 | 0.314 | 0.348 | 0.283 | 0.321 | 0.275 | 0.326 |
| Improve | – | – | 5.85% | 2.70% | 12.57% | 6.49% | – | – | 5.33% | 2.35% | 13.94% | 1.76% | – | – | 9.87% | 7.76% | 12.42% | 6.32% |
| Weather | 0.305 | 0.311 | 0.243 | 0.280 | 0.238 | 0.276 | 0.240 | 0.273 | 0.243 | 0.280 | 0.238 | 0.276 | 0.293 | 0.300 | 0.272 | 0.288 | 0.266 | 0.279 |
| Improve | – | – | 20.33% | 9.97% | 21.97% | 11.25% | – | – | -0.13% | -2.56% | 0.83% | -1.10% | – | – | 7.17% | 4.00% | 9.22% | 7.00% |

**Zero-shot Learning.** In Table 7, we delve into the performance of zero-shot learning tasks. Generally, the phenomenon of HD dominance, with TD consistently ranking second, remains evident across both intra-domain (e.g., ECL→ETTh1) and cross-domain (Traffic→ETTh1) tasks. As highlighted in Attraos Hu et al. (2024a), the underlying dynamical structures of time series display stable patterns, capturing the system's long-term evolutionary behaviors. Unlike numerical statistical information, which depends on specific datasets, ***the dynamical topological structures provide more fundamental insights with stronger generalization, leading to significant performance improvements.***

Table 7: Performance comparison for zero-shot forecasting with input 96 and forecasting length 96. The first and second results are highlighted in ▨ and ▨. Experiments are based on SimMTM (Dong et al., 2024).

| | Time-SSM | | +TD | | +HD | | PatchTST | | +TD | | +HD | | ModernTCN | | +TD | | +HD | |
|---|---|---|---|---|---|---|---|---|---|---|---|---|---|---|---|---|---|---|
| | MSE | MAE | MSE | MAE | MSE | MAE | MSE | MAE | MSE | MAE | MSE | MAE | MSE | MAE | MSE | MAE | MSE | MAE |
| ETTh2→ETTh1 | 0.548 | 0.488 | 0.517 | 0.467 | 0.524 | 0.477 | 0.527 | 0.482 | 0.532 | 0.489 | 0.509 | 0.465 | 0.488 | 0.461 | 0.511 | 0.488 | 0.476 | 0.460 |
| ETTm1→ETTh1 | 0.694 | 0.557 | 0.681 | 0.552 | 0.596 | 0.497 | 0.712 | 0.572 | 0.697 | 0.562 | 0.686 | 0.559 | 0.641 | 0.535 | 0.634 | 0.531 | 0.627 | 0.521 |
| ETTm2→ETTm1 | 0.610 | 0.484 | 0.607 | 0.494 | 0.564 | 0.486 | 0.626 | 0.474 | 0.575 | 0.470 | 0.520 | 0.456 | 0.650 | 0.516 | 0.624 | 0.497 | 0.616 | 0.485 |
| Weather→ETTh1 | 0.798 | 0.599 | 0.780 | 0.586 | 0.836 | 0.594 | 0.784 | 0.599 | 0.727 | 0.566 | 0.728 | 0.563 | 0.732 | 0.570 | 0.749 | 0.588 | 0.725 | 0.556 |
| ECL→ETTh1 | 0.412 | 0.410 | 0.400 | 0.405 | 0.396 | 0.400 | 0.439 | 0.437 | 0.401 | 0.405 | 0.398 | 0.401 | 0.481 | 0.466 | 0.439 | 0.437 | 0.448 | 0.434 |
| ECL→ETTm1 | 0.936 | 0.611 | 0.858 | 0.585 | 0.826 | 0.577 | 0.971 | 0.633 | 0.944 | 0.603 | 0.827 | 0.578 | 0.944 | 0.619 | 0.913 | 0.598 | 0.905 | 0.587 |
| Traffic→ETTh1 | 0.447 | 0.435 | 0.429 | 0.428 | 0.426 | 0.423 | 0.453 | 0.441 | 0.454 | 0.440 | 0.419 | 0.415 | 0.470 | 0.464 | 0.491 | 0.479 | 0.458 | 0.441 |

**Zero-shot Learning *w.r.t.* Input Length.** Table 8 provides an analysis of the impact of varying input lengths, highlighting key trends: (a) Overall, the HD method consistently maintains superior performance, with the TD method closely following. The occasional decline in TD performance may result from its sensitivity to the hyper-parameter in noisy real-world datasets, which can distort the dynamical structures. (b) Physics-guided embeddings exhibit more substantial improvements in the MAE metric, suggesting greater sensitivity to large outliers during forecasting. (c) Except for the ETTh2 dataset, both parameterized and physics-guided embeddings effectively leverage longer contextual information for cross-dataset prediction. Notably, the physics-guided embeddings show a more substantial performance enhancement, achieving an impressive improvement of over 50% at an input length of 720, which indicates ***their potential to become a new embedding paradigm.***

Table 8: Zero-shot learning results with various input lengths. The improved results are highlighted in ▨, and the decreased results are highlighted in ▨; improvements over 10% are highlighted in ▨.

| | Original-96 | | +TD | | +HD | | Original-336 | | +TD | | +HD | | Original-720 | | +TD | | +HD | |
|---|---|---|---|---|---|---|---|---|---|---|---|---|---|---|---|---|---|---|
| | MSE | MAE | MSE | MAE | MSE | MAE | MSE | MAE | MSE | MAE | MSE | MAE | MSE | MAE | MSE | MAE | MSE | MAE |
| ETTh2→ETTh1 | 0.527 | 0.482 | 0.532 | 0.489 | 0.509 | 0.465 | 0.711 | 0.574 | 0.507 | 0.480 | 0.482 | 0.453 | 0.959 | 0.660 | 0.471 | 0.468 | 0.449 | 0.446 |
| Improve | – | – | -0.96% | -1.46% | 3.37% | 3.36% | – | – | 28.66% | 16.48% | 32.23% | 21.18% | – | – | 50.91% | 29.06% | 53.17% | 32.40% |
| ETTm1→ETTh1 | 0.712 | 0.572 | 0.697 | 0.562 | 0.686 | 0.559 | 0.573 | 0.500 | 0.552 | 0.500 | 0.500 | 0.475 | 0.604 | 0.542 | 0.541 | 0.503 | 0.467 | 0.467 |
| Improve | – | – | 2.05% | 1.76% | 3.54% | 2.30% | – | – | 3.62% | -0.01% | 12.75% | 4.86% | – | – | 10.40% | 7.19% | 22.63% | 13.91% |
| ETTm2→ETTm1 | 0.626 | 0.474 | 0.575 | 0.470 | 0.520 | 0.456 | 0.460 | 0.435 | 0.386 | 0.406 | 0.424 | 0.409 | 0.412 | 0.427 | 0.409 | 0.419 | 0.398 | 0.400 |
| Improve | – | – | 8.07% | 1.04% | 16.83% | 3.97% | – | – | 16.08% | 6.67% | 7.95% | 5.98% | – | – | 0.70% | 1.84% | 3.46% | 6.22% |
| Weather→ETTh1 | 0.784 | 0.599 | 0.727 | 0.566 | 0.728 | 0.563 | 0.703 | 0.553 | 0.679 | 0.523 | 0.668 | 0.515 | 0.688 | 0.547 | 0.704 | 0.564 | 0.682 | 0.539 |
| Improve | – | – | 7.29% | 5.44% | 7.16% | 5.93% | – | – | 3.47% | 5.40% | 5.04% | 6.86% | – | – | -2.22% | -3.21% | 0.87% | -1.46% |
| ECL→ETTh1 | 0.439 | 0.437 | 0.401 | 0.405 | 0.398 | 0.401 | 0.409 | 0.418 | 0.387 | 0.404 | 0.386 | 0.403 | 0.371 | 0.405 | 0.371 | 0.403 | 0.364 | 0.397 |
| Improve | – | – | 8.52% | 7.40% | 9.34% | 8.30% | – | – | 5.42% | 3.46% | 5.57% | 3.77% | – | – | -0.02% | 0.49% | 1.98% | 2.14% |
| ECL→ETTm1 | 0.971 | 0.633 | 0.944 | 0.603 | 0.827 | 0.578 | 0.927 | 0.619 | 0.932 | 0.627 | 0.880 | 0.567 | 0.737 | 0.544 | 0.695 | 0.514 | 0.655 | 0.497 |
| Improve | – | – | 2.75% | 4.78% | 14.79% | 8.73% | – | – | -0.54% | -1.24% | 5.07% | 8.40% | – | – | 5.71% | 5.59% | 11.13% | 8.71% |
| Traffic→ETTh1 | 0.453 | 0.441 | 0.454 | 0.440 | 0.419 | 0.415 | 0.396 | 0.411 | 0.404 | 0.414 | 0.395 | 0.408 | 0.370 | 0.401 | 0.373 | 0.405 | 0.371 | 0.403 |
| Improve | – | – | -0.25% | 0.17% | 7.60% | 5.89% | – | – | -2.03% | -0.90% | 0.12% | 0.64% | – | – | 1.76% | 0.25% | 3.47% | 3.01% |

**Discussion About Scalability.** The remarkable improvements achieved by physics-guided embeddings in few-shot and zero-shot scenarios suggest their potential application in large-scale time series foundation models (LTSFM) (Jin et al., 2023). A crucial aspect of advancing towards physics-guided LTSFM is the necessity of the scaling laws. However, while physics-guided embeddings are available in model depth (layers) expansion, they are constrained by physical priors in model width (hidden dimensions), leading to significant constraints on memory capacity as the dataset size increases. One potential solution is to integrate physics-guided embeddings with a Mixture of Expert techniques (Cai et al., 2024). Diverse dynamical dimensions are established to enhance model representation and memory storage during the training stage, with an adaptive selection during the inference stage.

## 6 CONCLUSION & FUTURE WORK

Inspired by embedding theory, this paper demonstrates that the embedding layer in a deep time series model is an estimation of the underlying dynamics of the data. Based on this, we explore replacing parameterized embedding with numerical reconstruction techniques. Experiments show that physics-guided embeddings significantly improve performance across various tasks and backbones. In the future, we aim to advance physics-guided embeddings as a standard embedding technique for expert models and develop physics-guided time series foundation models (Liang et al., 2024).

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

## A  TECHNICAL BACKGROUND

### A.1  HOW TO DETERMINE THE PHYSICAL HYPER-PARAMETER

As mentioned earlier, the theoretical assurances of Takens' theorem falter under finite precision and noise, prompting the exploration of "optimal" embedding parameters. The notion of an "optimal" set suggests that embeddings vary in quality. Yet, assessing this quality necessitates a metric for comparison. Apart from the empirical selection methods mentioned in the paper, there are also other mainstream approaches available. Generally, these methods can be summarized in two broad categories or arguments: prediction-based and topological arguments.

- **Prediction-based** methods (Casdagli et al., 1991; Potapov, 1997) notions of embedding quality can be seen to be inspired by the application of embeddings in the context of time-series prediction. Fundamentally, good embeddings should enable better predictions. These methods generally try to maximize the amount of new information incorporated in each delay dimension with the aim that it will provide more information about the true system state and aid in time series prediction.
- **Topological** methods (Buzug & Pfister, 1992; Nichkawde, 2013) often concentrate on analyzing the attractor structure and the distribution of the manifold within its ambient space. Essentially, a well-structured embedding in terms of topology and geometry should aim to be adequately spread out and unfolded within its ambient setting. This concept of quality aligns with Casdagli's noise amplification arguments. Geometrically-based methods may encompass metrics like the fill factor and displacement from the diagonal. In essence, the considerations for determining the optimal lag and embedding dimension for time delay embedding can be encapsulated by the notions of irrelevance and redundancy.

### A.2  DYNAMICAL ENCODER

In the field of machine learning, particularly in the realm of dynamical systems modeling, articles on chaotic dynamical systems primarily focus on employing recurrent neural networks (RNNs) (Mikhaeil et al., 2022; Hess et al., 2023) and state-space models (Hu et al., 2024a; Alonso et al., 2024) for modeling, relying on the autoregressive nature of models to capture underlying dynamics. Additionally, some studies are dedicated to reservoir computing (Yan et al., 2024), simulating the problems sensitive to initial values of partial differential equations by maintaining a random vector reservoir. Furthermore, Neural ODEs (Li et al., 2020; Gupta et al., 2021) and some Physics-Informed Neural Networks (PINNs) (Raissi et al., 2019) attempt to uncover underlying patterns in data through a combination of data-driven and physics-constrained approaches. Increasingly, research indicates that deep learning models, i,e., Transformers (Hang et al., 2024) can also achieve impressive results.

## B  PROOFS

**Proposition B.1.** *The embedding, which uses a shared linear matrix, is an integral transformation $h(t) = \int_{-\infty}^{t} x(s)\phi(t,s)\mathrm{d}\mu(s)$ with limited time-invariant measure $\mu$ and polynomial basis $\phi$.*

*Proof.* This perspective has been extensively discussed in numerous relevant literature (Gu et al., 2020; 2022; Hu et al., 2024c;b), where both patch operations and convolutional neural networks are seen as a parameterized continuous convolution process under a uniform and finite measure window, akin to a polynomial basis function projection. The Hippo theory (Gu et al., 2020) provides a detailed theoretical framework for this. Various extensions can be derived based on different basis functions and measure windows; for instance, trigonometric basis functions lead to Fourier transforms, piecewise polynomial bases result in wavelet transforms, and exponential decay bases yield the recent deep state space model S4 Gu et al. (2021).  □

**Proposition B.2.** *For the full-rank embedding matrix, the embedding process is a similarity transformation that maintains the original dynamical properties (system eigenvalues).*

*Proof.* Let the underlying nonlinear time-variant dynamics is $\dot{x} = g(x(t), t)$, the dynamics for the (patched) hidden state $h$ is:

$$\boldsymbol{h}_{i+1} = \boldsymbol{W}\bar{J}\boldsymbol{W}^{-1}\boldsymbol{h}_i, \qquad \bar{J} = \frac{1}{P}\sum_{i=1}^{P} J\left(x\left(t_i\right)\right), \qquad J(x) = \frac{\partial g(x)}{\partial x}, \qquad (4)$$

which can be regarded as a ***linearization for non-linear time series***. The patch operation averages the Jacobian matrix $J$ that characterizes the system dynamics. As the patch length increases, the embedding will discard more nonlinear features of the data. For a full-rank matrix $\boldsymbol{W}$, the transformation $\boldsymbol{W}\bar{J}\boldsymbol{W}^{-1}$, known as a Similarity Transformation, where $\boldsymbol{W}\bar{J}\boldsymbol{W}^{-1}$ is unitarily equivalent to $\bar{J}$, preserves the underlying dynamical properties, serving as a mere space coordinate transformation. □

**Lemma B.3.** *A continuous function $f$ is K-Lipschitz when $\|f(x_1) - f(x_2)\| \leq K\|x_1 - x_2\|$, then:*

*(1) The state space model with the negative diagonal matrix $\boldsymbol{A}$ and normalization layers is 1-Lipschitz.*

*(2) The fully connected and convolution neural network with normalization layers is 1-Lipschitz.*

*(3) The standard dot-product attention is not Lipschitz. The $L_2$ attention is bounded Lipschitz.*

*Proof.* The first part can be found with detailed proof in **Lemma 2.8** of Time-SSM (Hu et al., 2024b), and matrices $\boldsymbol{A}$ with negative diagonal eigenvalues can also be explained using left half-plane control theory. Descriptions of the second and third parts can be found in (Kim et al., 2021). According to this theorem, SSMs, CNNs, and MLPs (although not suitable for our physics-guided embeddings) can be used to stabilize the modeling of dynamical structures to preserve dynamical characteristics, while the Transformer architecture may potentially disrupt underlying dynamics during modeling. However, some recent articles have shown promising results using the Transformer architecture in modeling PDE dynamical systems (Hang et al., 2024; Zhang & Gilpin, 2024), warranting further exploration in the future. □

**Proposition B.4.** *The Lyapunov exponents $\lambda_m = \lim_{t \to \infty} \frac{1}{t}\ln\sigma_m(t)$ of the system attractors are the mean logarithmic growth rates of the principal axes lengths of the ellipsoidal feature space.*

*Proof.* This is a standard theory in nonlinear dynamical systems, with detailed explanations available in **Section 1.2** of the relevant literature (Skokos et al., 2016). □

## C  EXPERIMENTS

### C.1  ENCODER BACKBONE

In this paper, we have selected state-of-the-art models based on the CNN, Transformer, and SSM architectures as the backbone encoders. The specific details are as follows.

- **Modern-TCN** (Luo & Wang, 2024) is a pure convolutional architecture that incorporates both upsampling, downsampling techniques, and patching methods to stack models that separately capture temporal and channel correlations.
- **PatchTST** (Nie et al., 2022) is the first transformer architecture to introduce chunking operations, employing a channel-independent strategy to apply the same backbone model to each time variable. It continues to maintain state-of-the-art performance in many tasks to this day.
- **TimeSSM** (Hu et al., 2024b) is a recent model architecture that applies the SSM kernel, typically utilizing patching operation and channel-independent modeling strategies, particularly excelling in prediction tasks with outstanding performance.

### C.2  DATASETS

We perform experiments on 8 authentic datasets to assess our model's performance, with detailed information provided in Table 9. The Dimension signifies the variable count in each dataset. Dataset Size indicates the total time points in the train, validation, and test splits. Forecasting Length specifies the future time points for prediction, with four prediction settings per dataset. Frequency represents the time point sampling interval. To elaborate:

- **ETT** dataset (Zhou et al., 2021) encompasses 7 electricity transformer factors spanning from July 2016 to July 2018. We utilize four subsets: ETTh1 and ETTh2 are hourly recorded, while ETTm1 and ETTm2 are recorded every 15 minutes.
- **Exchange** (Wu et al., 2021) compiles daily exchange rate panel data from 8 countries between 1990 and 2016.
- **Weather** (Wu et al., 2021) integrates 21 meteorological factors recorded every 10 minutes from the Weather Station of the Max Planck Bio-geochemistry Institute in 2020.
- **Electricity** (Wu et al., 2021) records the hourly electricity consumption data of 321 clients.
- **Traffic** (Wu et al., 2021) collects hourly road occupancy rates measured by 862 sensors of San Francisco Bay area freeways from January 2015 to December 2016. The train, validation, and test datasets are strictly divided according to chronological order to make sure there are no data leakage.

Table 9: Detailed dataset descriptions.

| Dataset | Dimension | Forecasting Length | Dataset Size | Information (Frequency) |
|---------|-----------|--------------------|--------------|-------------------------|
| ETTm1 | 7 | {96, 192, 336, 720} | (34369, 11425, 11425) | Electricity (15 min) |
| ETTh1 | 7 | {96, 192, 336, 720} | (8445, 2785, 2785) | Electricity (Hourly) |
| ETTm2 | 7 | {96, 192, 336, 720} | (34369, 11425, 11425) | Electricity (15 min) |
| ETTh2 | 7 | {96, 192, 336, 720} | (8545, 2881, 2881) | Electricity (Hourly) |
| Exchange | 8 | {96, 192, 336, 720} | (5120, 665, 1422) | Exchange rate (Daily) |
| Weather | 21 | {96, 192, 336, 720} | (36696, 5175, 10440) | Weather (10 min) |
| Electricity | 321 | {96, 192, 336, 720} | (18221, 2537, 5165) | Electricity (Hourly) |
| Traffic | 862 | {96, 192, 336, 720} | (12089, 1661, 3413) | Transportation (Hourly) |

## C.3 EXPERIMENT SETTING

All experiments are conducted on the NVIDIA A6000-48G GPUs. The Adam optimizer is chosen. To ensure a fair and comprehensive comparison of the superiority of our proposed method, we conduct a complete set of experiments on the Time Series Library architecture. Throughout the experimental process, we ensure consistency in the application of physics-guided embeddings and parameterized embeddings, maintaining the same hyper-parameters in both the model architecture and experimental procedures. Specifically, the number of model layers, patch length, and stride are set based on the original paper's configurations, with a learning rate of 0.0001 and a hidden dimension of 256.

## C.4 MORE VISUALIZATION

In Figure 12, we present the spectral diagram of embeddings obtained through more parameterized embeddings and physics-guided embeddings.

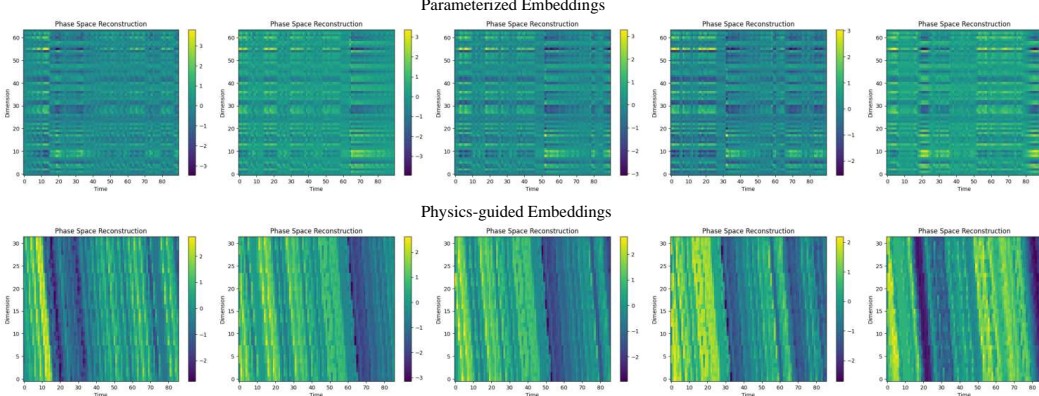

Figure 12: More visualizations for parameterized embedding and physics-guided embedding

## C.5 FULL RESULTS

We present the full experiment results in the following tables.

Table 10: Full results for the anomaly detection task. The P, R, and F1 represent the precision, recall, and F1-score (%), respectively. A higher value of P, R, and F1 indicates a better performance.

| Datasets | PSM | | | MSL | | | SMAP | | | SMD | | | SWAT | | | Avg |
|---|---|---|---|---|---|---|---|---|---|---|---|---|---|---|---|---|
| Metrics | R | F1 | P | R | F1 | P | R | F1 | P | R | F1 | P | R | F1 | P | |
| Time-SSM | 0.936 | 0.959 | 0.983 | 0.725 | 0.801 | 0.895 | 0.602 | 0.726 | 0.913 | 0.909 | 0.844 | 0.788 | 0.913 | 0.917 | 0.921 | 0.849 |
| TD-Emb | 0.943 | 0.962 | 0.989 | 0.727 | 0.805 | 0.902 | 0.603 | 0.725 | 0.915 | 0.908 | 0.842 | 0.787 | 0.919 | 0.925 | 0.926 | 0.852 |
| HD-Emb | 0.939 | 0.957 | 0.981 | 0.732 | 0.808 | 0.900 | 0.606 | 0.731 | 0.920 | 0.909 | 0.845 | 0.791 | 0.912 | 0.927 | 0.926 | 0.854 |
| PatchTST | 0.950 | 0.969 | 0.989 | 0.713 | 0.790 | 0.886 | 0.536 | 0.673 | 0.902 | 0.861 | 0.810 | 0.764 | 0.827 | 0.868 | 0.913 | 0.822 |
| TD-Emb | 0.936 | 0.959 | 0.983 | 0.714 | 0.784 | 0.881 | 0.532 | 0.670 | 0.902 | 0.845 | 0.802 | 0.764 | 0.930 | 0.926 | 0.923 | 0.828 |
| HD-Emb | 0.941 | 0.962 | 0.984 | 0.715 | 0.783 | 0.881 | 0.536 | 0.673 | 0.902 | 0.854 | 0.807 | 0.764 | 0.921 | 0.922 | 0.922 | 0.829 |
| ModernTCN | 0.945 | 0.965 | 0.986 | 0.749 | 0.816 | 0.896 | 0.558 | 0.691 | 0.908 | 0.816 | 0.844 | 0.874 | 0.903 | 0.930 | 0.958 | 0.849 |
| TD-Emb | 0.944 | 0.966 | 0.987 | 0.745 | 0.813 | 0.892 | 0.563 | 0.696 | 0.913 | 0.819 | 0.847 | 0.876 | 0.900 | 0.927 | 0.956 | 0.850 |
| HD-Emb | 0.947 | 0.965 | 0.988 | 0.754 | 0.822 | 0.891 | 0.560 | 0.693 | 0.909 | 0.812 | 0.841 | 0.872 | 0.914 | 0.944 | 0.969 | 0.853 |

Table 11: Multivariate long-term series forecasting results with input length are{96, 336, 720} on PatchTST.

| Model | | PatchTST-336-ori | | PatchTST-336-TD | | PatchTST-336-HD | | PatchTST-720-ori | | PatchTST-720-TD | | PatchTST-720-HD | |
|---|---|---|---|---|---|---|---|---|---|---|---|---|---|
| Efficiency | | MSE | MAE | MSE | MAE | MSE | MAE | MSE | MAE | MSE | MAE | MSE | MAE |
| ETTh1 | 96 | 0.376 | 0.407 | 0.381 | 0.405 | 0.382 | 0.400 | 0.392 | 0.428 | 0.379 | 0.414 | 0.370 | 0.404 |
| | 192 | 0.424 | 0.441 | 0.424 | 0.430 | 0.413 | 0.420 | 0.431 | 0.449 | 0.417 | 0.437 | 0.426 | 0.441 |
| | 336 | 0.423 | 0.441 | 0.431 | 0.432 | 0.414 | 0.428 | 0.469 | 0.474 | 0.424 | 0.441 | 0.488 | 0.492 |
| | 720 | 0.455 | 0.474 | 0.432 | 0.460 | 0.444 | 0.442 | 0.508 | 0.504 | 0.465 | 0.481 | 0.678 | 0.578 |
| | AVG | 0.420 | 0.441 | 0.417 | 0.432 | 0.413 | 0.422 | 0.481 | 0.483 | 0.421 | 0.443 | 0.490 | 0.479 |
| ETTh2 | 96 | 0.295 | 0.354 | 0.286 | 0.351 | 0.298 | 0.354 | 0.313 | 0.372 | 0.293 | 0.358 | 0.293 | 0.354 |
| | 192 | 0.351 | 0.424 | 0.347 | 0.390 | 0.363 | 0.391 | 0.393 | 0.418 | 0.348 | 0.396 | 0.347 | 0.390 |
| | 336 | 0.375 | 0.414 | 0.373 | 0.407 | 0.374 | 0.405 | 0.502 | 0.471 | 0.378 | 0.413 | 0.376 | 0.424 |
| | 720 | 0.410 | 0.469 | 0.410 | 0.441 | 0.416 | 0.450 | 0.548 | 0.527 | 0.412 | 0.454 | 0.428 | 0.463 |
| | AVG | 0.358 | 0.415 | 0.354 | 0.397 | 0.363 | 0.400 | 0.439 | 0.447 | 0.357 | 0.405 | 0.361 | 0.408 |
| ETTm2 | 96 | 0.167 | 0.256 | 0.173 | 0.264 | 0.170 | 0.258 | 0.178 | 0.274 | 0.185 | 0.277 | 0.172 | 0.262 |
| | 192 | 0.225 | 0.299 | 0.232 | 0.302 | 0.221 | 0.292 | 0.240 | 0.312 | 0.256 | 0.321 | 0.241 | 0.310 |
| | 336 | 0.281 | 0.337 | 0.276 | 0.332 | 0.282 | 0.338 | 0.293 | 0.347 | 0.287 | 0.347 | 0.292 | 0.345 |
| | 720 | 0.380 | 0.402 | 0.368 | 0.388 | 0.365 | 0.388 | 0.407 | 0.421 | 0.370 | 0.401 | 0.386 | 0.408 |
| | AVG | 0.263 | 0.323 | 0.262 | 0.321 | 0.259 | 0.319 | 0.280 | 0.339 | 0.275 | 0.337 | 0.273 | 0.331 |
| Weather | 96 | 0.154 | 0.204 | 0.152 | 0.202 | 0.154 | 0.204 | 0.152 | 0.208 | 0.149 | 0.201 | 0.148 | 0.199 |
| | 192 | 0.201 | 0.248 | 0.197 | 0.244 | 0.198 | 0.244 | 0.204 | 0.256 | 0.194 | 0.245 | 0.190 | 0.239 |
| | 336 | 0.248 | 0.284 | 0.249 | 0.284 | 0.248 | 0.281 | 0.249 | 0.291 | 0.250 | 0.290 | 0.239 | 0.278 |
| | 720 | 0.330 | 0.342 | 0.326 | 0.336 | 0.320 | 0.332 | 0.316 | 0.338 | 0.317 | 0.333 | 0.310 | 0.325 |
| | AVG | 0.233 | 0.270 | 0.231 | 0.267 | 0.230 | 0.265 | 0.231 | 0.273 | 0.227 | 0.267 | 0.222 | 0.260 |

Table 12: Few-shot results: input length is 336, prediction horizons {96, 192, 336, 720}.

| Model | | | | Time-SSM | | PatchTST | | ModernTCN | |
|---|---|---|---|---|---|---|---|---|---|
| Metric | | | | MSE | MAE | MSE | MAE | MSE | MAE |
| ETTh1 | Original | | 96 | 0.546 | 0.508 | 0.452 | 0.460 | 0.457 | 0.460 |
| | | | 192 | 0.683 | 0.573 | 0.527 | 0.507 | 0.527 | 0.498 |
| | | | 336 | 0.846 | 0.654 | 0.770 | 0.629 | 0.637 | 0.544 |
| | | | 720 | 0.977 | 0.735 | 0.908 | 0.671 | 0.707 | 0.596 |
| | TD-Emb | | 96 | 0.535 | 0.500 | 0.477 | 0.470 | 0.421 | 0.442 |
| | | | 192 | 0.677 | 0.567 | 0.516 | 0.486 | 0.481 | 0.469 |
| | | | 336 | 0.840 | 0.643 | 0.539 | 0.505 | 0.600 | 0.531 |
| | | | 720 | 1.527 | 0.892 | 0.809 | 0.634 | 0.667 | 0.575 |
| | HD-Emb | | 96 | 0.530 | 0.497 | 0.470 | 0.456 | 0.417 | 0.440 |
| | | | 192 | 0.614 | 0.543 | 0.515 | 0.482 | 0.475 | 0.452 |
| | | | 336 | 0.760 | 0.619 | 0.545 | 0.513 | 0.598 | 0.533 |
| | | | 720 | 1.310 | 0.832 | 0.739 | 0.609 | 0.643 | 0.564 |
| ETTh2 | Original | | 96 | 0.377 | 0.408 | 0.351 | 0.386 | 0.315 | 0.362 |
| | | | 192 | 0.464 | 0.454 | 0.427 | 0.429 | 0.400 | 0.408 |
| | | | 336 | 0.559 | 0.514 | 0.463 | 0.460 | 0.391 | 0.418 |
| | | | 720 | 0.660 | 0.564 | 0.554 | 0.519 | 0.456 | 0.467 |
| | TD-Emb | | 96 | 0.370 | 0.406 | 0.332 | 0.377 | 0.319 | 0.365 |
| | | | 192 | 0.461 | 0.454 | 0.392 | 0.411 | 0.422 | 0.401 |
| | | | 336 | 0.509 | 0.493 | 0.416 | 0.438 | 0.388 | 0.437 |
| | | | 720 | 0.652 | 0.562 | 0.510 | 0.498 | 0.459 | 0.447 |
| | HD-Emb | | 96 | 0.350 | 0.393 | 0.318 | 0.367 | 0.306 | 0.365 |
| | | | 192 | 0.423 | 0.439 | 0.399 | 0.423 | 0.401 | 0.416 |
| | | | 336 | 0.577 | 0.531 | 0.400 | 0.434 | 0.391 | 0.414 |
| | | | 720 | 0.635 | 0.561 | 0.474 | 0.481 | 0.439 | 0.453 |
| ETTm2 | Original | | 96 | 0.224 | 0.300 | 0.196 | 0.275 | 0.233 | 0.297 |
| | | | 192 | 0.285 | 0.342 | 0.257 | 0.314 | 0.291 | 0.333 |
| | | | 336 | 0.361 | 0.386 | 0.308 | 0.349 | 0.325 | 0.357 |
| | | | 720 | 0.496 | 0.453 | 0.440 | 0.423 | 0.406 | 0.405 |
| | TD-Emb | | 96 | 0.211 | 0.293 | 0.202 | 0.282 | 0.207 | 0.277 |
| | | | 192 | 0.284 | 0.339 | 0.252 | 0.313 | 0.256 | 0.316 |
| | | | 336 | 0.353 | 0.381 | 0.301 | 0.342 | 0.303 | 0.331 |
| | | | 720 | 0.439 | 0.426 | 0.381 | 0.392 | 0.366 | 0.381 |
| | HD-Emb | | 96 | 0.208 | 0.292 | 0.197 | 0.279 | 0.201 | 0.280 |
| | | | 192 | 0.259 | 0.322 | 0.248 | 0.312 | 0.249 | 0.312 |
| | | | 336 | 0.315 | 0.360 | 0.298 | 0.343 | 0.283 | 0.327 |
| | | | 720 | 0.412 | 0.412 | 0.395 | 0.400 | 0.368 | 0.385 |
| Weather | Original | | 96 | 0.197 | 0.229 | 0.161 | 0.208 | 0.187 | 0.228 |
| | | | 192 | 0.257 | 0.285 | 0.206 | 0.251 | 0.281 | 0.289 |
| | | | 336 | 0.323 | 0.328 | 0.259 | 0.290 | 0.335 | 0.325 |
| | | | 720 | 0.444 | 0.402 | 0.334 | 0.345 | 0.369 | 0.360 |
| | TD-Emb | | 96 | 0.163 | 0.220 | 0.163 | 0.220 | 0.179 | 0.215 |
| | | | 192 | 0.209 | 0.259 | 0.209 | 0.259 | 0.256 | 0.277 |
| | | | 336 | 0.263 | 0.297 | 0.263 | 0.297 | 0.317 | 0.317 |
| | | | 720 | 0.335 | 0.346 | 0.335 | 0.346 | 0.337 | 0.344 |
| | HD-Emb | | 96 | 0.160 | 0.216 | 0.160 | 0.216 | 0.172 | 0.211 |
| | | | 192 | 0.203 | 0.252 | 0.203 | 0.252 | 0.252 | 0.263 |
| | | | 336 | 0.258 | 0.293 | 0.258 | 0.293 | 0.310 | 0.308 |
| | | | 720 | 0.332 | 0.345 | 0.332 | 0.345 | 0.330 | 0.334 |

Table 13: Multivariate long-term series forecasting results: input 96, prediction horizons {96, 192, 336, 720}.

| Model | | | Time-SSM | | PatchTST | | ModernTCN | |
|---|---|---|---|---|---|---|---|---|
| Metric | | | MSE | MAE | MSE | MAE | MSE | MAE |
| ETTh1 | Original | 96 | 0.378 | 0.397 | 0.378 | 0.398 | 0.389 | 0.397 |
| | | 192 | 0.431 | 0.432 | 0.425 | 0.432 | 0.437 | 0.426 |
| | | 336 | 0.472 | 0.450 | 0.470 | 0.458 | 0.477 | 0.442 |
| | | 720 | 0.475 | 0.473 | 0.525 | 0.507 | 0.478 | 0.464 |
| | | AVG | 0.439 | 0.438 | 0.450 | 0.449 | 0.445 | 0.432 |
| | TD-Emb | 96 | 0.377 | 0.397 | 0.375 | 0.397 | 0.388 | 0.395 |
| | | 192 | 0.429 | 0.427 | 0.423 | 0.430 | 0.434 | 0.424 |
| | | 336 | 0.479 | 0.454 | 0.469 | 0.452 | 0.471 | 0.438 |
| | | 720 | 0.459 | 0.465 | 0.484 | 0.479 | 0.470 | 0.436 |
| | | AVG | 0.436 | 0.436 | 0.438 | 0.440 | 0.441 | 0.423 |
| | HD-Emb | 96 | 0.377 | 0.399 | 0.372 | 0.395 | 0.383 | 0.394 |
| | | 192 | 0.424 | 0.424 | 0.420 | 0.427 | 0.431 | 0.423 |
| | | 336 | 0.470 | 0.457 | 0.472 | 0.448 | 0.475 | 0.461 |
| | | 720 | 0.458 | 0.460 | 0.485 | 0.479 | 0.472 | 0.437 |
| | | AVG | 0.432 | 0.435 | 0.437 | 0.437 | 0.440 | 0.429 |
| ETTh2 | Original | 96 | 0.291 | 0.342 | 0.291 | 0.346 | 0.292 | 0.340 |
| | | 192 | 0.374 | 0.383 | 0.378 | 0.404 | 0.378 | 0.394 |
| | | 336 | 0.421 | 0.431 | 0.425 | 0.440 | 0.427 | 0.433 |
| | | 720 | 0.430 | 0.448 | 0.436 | 0.454 | 0.433 | 0.448 |
| | | AVG | 0.379 | 0.401 | 0.382 | 0.411 | 0.382 | 0.404 |
| | TD-Emb | 96 | 0.288 | 0.340 | 0.287 | 0.343 | 0.289 | 0.339 |
| | | 192 | 0.375 | 0.382 | 0.374 | 0.398 | 0.375 | 0.392 |
| | | 336 | 0.417 | 0.429 | 0.417 | 0.423 | 0.424 | 0.428 |
| | | 720 | 0.425 | 0.445 | 0.425 | 0.440 | 0.422 | 0.439 |
| | | AVG | 0.376 | 0.399 | 0.376 | 0.401 | 0.378 | 0.400 |
| | HD-Emb | 96 | 0.289 | 0.342 | 0.287 | 0.338 | 0.288 | 0.340 |
| | | 192 | 0.372 | 0.382 | 0.374 | 0.393 | 0.373 | 0.391 |
| | | 336 | 0.419 | 0.427 | 0.416 | 0.427 | 0.419 | 0.426 |
| | | 720 | 0.416 | 0.439 | 0.418 | 0.435 | 0.425 | 0.441 |
| | | AVG | 0.374 | 0.398 | 0.374 | 0.398 | 0.376 | 0.400 |
| ETTm1 | Original | 96 | 0.336 | 0.371 | 0.324 | 0.364 | 0.318 | 0.360 |
| | | 192 | 0.369 | 0.389 | 0.372 | 0.392 | 0.363 | 0.390 |
| | | 336 | 0.397 | 0.411 | 0.399 | 0.408 | 0.399 | 0.409 |
| | | 720 | 0.455 | 0.443 | 0.458 | 0.445 | 0.463 | 0.446 |
| | | AVG | 0.389 | 0.403 | 0.388 | 0.402 | 0.386 | 0.401 |
| | TD-Emb | 96 | 0.336 | 0.372 | 0.322 | 0.363 | 0.319 | 0.361 |
| | | 192 | 0.367 | 0.388 | 0.365 | 0.381 | 0.365 | 0.392 |
| | | 336 | 0.394 | 0.408 | 0.397 | 0.404 | 0.397 | 0.407 |
| | | 720 | 0.451 | 0.439 | 0.456 | 0.435 | 0.467 | 0.449 |
| | | AVG | 0.387 | 0.402 | 0.385 | 0.396 | 0.387 | 0.402 |
| | HD-Emb | 96 | 0.332 | 0.368 | 0.320 | 0.361 | 0.318 | 0.359 |
| | | 192 | 0.371 | 0.391 | 0.362 | 0.380 | 0.361 | 0.388 |
| | | 336 | 0.393 | 0.405 | 0.388 | 0.398 | 0.393 | 0.406 |
| | | 720 | 0.441 | 0.435 | 0.440 | 0.431 | 0.455 | 0.438 |
| | | AVG | 0.384 | 0.400 | 0.378 | 0.393 | 0.382 | 0.398 |
| ETTm2 | Original | 96 | 0.176 | 0.260 | 0.177 | 0.263 | 0.172 | 0.255 |
| | | 192 | 0.247 | 0.309 | 0.250 | 0.310 | 0.243 | 0.303 |
| | | 336 | 0.305 | 0.344 | 0.311 | 0.349 | 0.310 | 0.345 |
| | | 720 | 0.408 | 0.407 | 0.423 | 0.415 | 0.415 | 0.405 |
| | | AVG | 0.284 | 0.330 | 0.291 | 0.334 | 0.285 | 0.327 |
| | TD-Emb | 96 | 0.177 | 0.261 | 0.177 | 0.261 | 0.175 | 0.254 |
| | | 192 | 0.246 | 0.306 | 0.241 | 0.303 | 0.245 | 0.305 |
| | | 336 | 0.305 | 0.343 | 0.302 | 0.347 | 0.317 | 0.349 |
| | | 720 | 0.411 | 0.409 | 0.402 | 0.401 | 0.422 | 0.411 |
| | | AVG | 0.285 | 0.330 | 0.281 | 0.328 | 0.290 | 0.330 |
| | HD-Emb | 96 | 0.177 | 0.261 | 0.175 | 0.262 | 0.174 | 0.252 |
| | | 192 | 0.243 | 0.305 | 0.241 | 0.306 | 0.237 | 0.299 |
| | | 336 | 0.301 | 0.341 | 0.305 | 0.348 | 0.314 | 0.348 |
| | | 720 | 0.406 | 0.406 | 0.412 | 0.404 | 0.417 | 0.404 |
| | | AVG | 0.282 | 0.328 | 0.283 | 0.330 | 0.286 | 0.326 |
| Weather | Original | 96 | 0.171 | 0.213 | 0.175 | 0.218 | 0.158 | 0.204 |
| | | 192 | 0.217 | 0.256 | 0.221 | 0.256 | 0.207 | 0.251 |
| | | 336 | 0.276 | 0.297 | 0.280 | 0.298 | 0.265 | 0.292 |
| | | 720 | 0.353 | 0.348 | 0.356 | 0.349 | 0.341 | 0.344 |
| | | AVG | 0.254 | 0.279 | 0.258 | 0.280 | 0.243 | 0.273 |
| | TD-Emb | 96 | 0.166 | 0.210 | 0.172 | 0.216 | 0.158 | 0.215 |
| | | 192 | 0.215 | 0.254 | 0.220 | 0.259 | 0.211 | 0.255 |
| | | 336 | 0.279 | 0.299 | 0.271 | 0.295 | 0.264 | 0.287 |
| | | 720 | 0.345 | 0.342 | 0.349 | 0.340 | 0.335 | 0.343 |
| | | AVG | 0.251 | 0.276 | 0.253 | 0.278 | 0.242 | 0.275 |
| | HD-Emb | 96 | 0.167 | 0.211 | 0.176 | 0.221 | 0.153 | 0.200 |
| | | 192 | 0.218 | 0.256 | 0.226 | 0.263 | 0.204 | 0.247 |
| | | 336 | 0.274 | 0.295 | 0.281 | 0.297 | 0.261 | 0.285 |
| | | 720 | 0.355 | 0.350 | 0.348 | 0.345 | 0.332 | 0.340 |
| | | AVG | 0.254 | 0.278 | 0.258 | 0.282 | 0.238 | 0.268 |
| Electricity | Original | 96 | 0.177 | 0.266 | 0.180 | 0.273 | 0.198 | 0.275 |
| | | 192 | 0.185 | 0.274 | 0.187 | 0.280 | 0.198 | 0.278 |
| | | 336 | 0.202 | 0.291 | 0.204 | 0.296 | 0.212 | 0.293 |
| | | 720 | 0.249 | 0.326 | 0.246 | 0.328 | 0.254 | 0.326 |
| | | AVG | 0.203 | 0.289 | 0.204 | 0.294 | 0.215 | 0.293 |
| | TD-Emb | 96 | 0.176 | 0.265 | 0.188 | 0.281 | 0.201 | 0.279 |
| | | 192 | 0.188 | 0.276 | 0.201 | 0.295 | 0.199 | 0.291 |
| | | 336 | 0.201 | 0.289 | 0.220 | 0.314 | 0.208 | 0.290 |
| | | 720 | 0.245 | 0.323 | 0.248 | 0.341 | 0.258 | 0.328 |
| | | AVG | 0.203 | 0.288 | 0.214 | 0.308 | 0.217 | 0.297 |
| | HD-Emb | 96 | 0.178 | 0.266 | 0.185 | 0.276 | 0.197 | 0.274 |
| | | 192 | 0.183 | 0.273 | 0.182 | 0.283 | 0.199 | 0.276 |
| | | 336 | 0.201 | 0.294 | 0.201 | 0.303 | 0.205 | 0.288 |
| | | 720 | 0.242 | 0.322 | 0.249 | 0.333 | 0.251 | 0.324 |
| | | AVG | 0.201 | 0.289 | 0.204 | 0.299 | 0.213 | 0.291 |
| Exchange | Original | 96 | 0.087 | 0.205 | 0.097 | 0.216 | 0.102 | 0.227 |
| | | 192 | 0.181 | 0.304 | 0.182 | 0.304 | 0.202 | 0.322 |
| | | 336 | 0.340 | 0.422 | 0.342 | 0.426 | 0.354 | 0.431 |
| | | 720 | 0.861 | 0.698 | 0.951 | 0.731 | 0.915 | 0.723 |
| | | AVG | 0.367 | 0.407 | 0.393 | 0.419 | 0.393 | 0.425 |
| | TD-Emb | 96 | 0.082 | 0.201 | 0.083 | 0.202 | 0.089 | 0.208 |
| | | 192 | 0.175 | 0.299 | 0.177 | 0.299 | 0.190 | 0.311 |
| | | 336 | 0.332 | 0.418 | 0.329 | 0.416 | 0.340 | 0.422 |
| | | 720 | 0.840 | 0.688 | 0.841 | 0.695 | 0.874 | 0.705 |
| | | AVG | 0.357 | 0.402 | 0.358 | 0.403 | 0.373 | 0.412 |
| | HD-Emb | 96 | 0.082 | 0.202 | 0.087 | 0.207 | 0.091 | 0.210 |
| | | 192 | 0.172 | 0.296 | 0.180 | 0.301 | 0.194 | 0.316 |
| | | 336 | 0.334 | 0.417 | 0.330 | 0.416 | 0.344 | 0.426 |
| | | 720 | 0.833 | 0.685 | 0.847 | 0.692 | 0.880 | 0.709 |
| | | AVG | 0.355 | 0.400 | 0.361 | 0.404 | 0.377 | 0.415 |

