# OpenReview forum: "Toward Physics-guided Time Series Embedding"
_ICLR.cc/2025/Conference — Submitted to ICLR 2025_

### Official Review · Reviewer_ksH4 · 2024-11-01

**Soundness:** 3
**Presentation:** 2
**Contribution:** 2
**Rating:** 5
**Confidence:** 3

**Summary:**

The authors introduce the concept of Embedding Duality which states that the hidden state representation in a deep time series model is equivalent to the underlying dynamical system structure of the data. They argue that parameterized embeddings in deep time series models are essentially linear estimations of the underlying nonlinear dynamics.
The paper makes the following contributions:

- Theoretical justification for Embedding Duality: The authors provide proofs to support the idea and indications of Embedding Duality.
- Empirical evidence for Embedding Duality: The authors provide several empirical results to support their theory, including visualizations of embedding representations, analysis of the dimension scaling law, and the impact of causal-directional modeling.
- Improved performance in expert models: The authors demonstrate that physics-guided embeddings significantly improve the performance of existing deep time series models in various expert tasks, including forecasting, classification, imputation, and anomaly detection. They evaluate different physics-guided embedding techniques, including Time Delay, Principal Component, High-order Derivatives, and Integral-Differential methods. They also show that physics-guided embeddings lead to a reduction in the number of parameters, increased efficiency, and improved robustness.
- Improved performance in foundation models: The authors show that physics-guided embeddings also improve the performance of foundation models in few-shot and zero-shot learning scenarios. They argue that this is because the dynamical structures captured by physics-guided embeddings are more generalizable than numerical statistical information.

After the discussion, my concern that the core concept of this paper, embedding duality, lacks formal definition still remains. I have decided to retain the score.

**Strengths:**

- Theoretical justification
- Plug-and-play framework
- Empirically competitive performance

**Weaknesses:**

- The description of Embedding Duality is not adequately clear. The terms such as "equivalent" and "dynamical system structure" lack of definitions. Suggestion: provide formal definitions for key terms like "equivalent" and "dynamical system structure" in the context of Embedding Duality theory.
- It is not very clear which parts were implemented or invented by the authors in their embedding-encoder-decoder structure. Suggestion: explicitly state which components of the embedding-encoder-decoder structure are novel contributions versus existing methods.
- It needs elaboration why and how the existing methods did not work as well as the proposed if those embedding methods in sec 4.1 have already existed. Suggestion: provide a detailed comparison between their implementation of these embedding methods and previous implementations, highlighting any key differences that may account for the improved performance.

**Questions:**

- Does the term "physics-guided" mean (linear) dynamical systems?
- Does the proposed approach work better because the authors implemented a linear dynamical system or just because the existing embedding methods implicitly have a DS in them?
- What is Y in the problem statement? Embedding vector?
- What does encoder do if embedding is mapping observation to some representation?
- What is the difference between the embedding in this work and the latent state in a SSM?
- (minor) It is best not to put code links in the abstract.

---

### Official Review · Reviewer_LWUw · 2024-11-02

**Soundness:** 2
**Presentation:** 1
**Contribution:** 2
**Rating:** 5
**Confidence:** 3

**Summary:**

The authors propose and empirically evaluate new patching/embedding strategies (time delay, principal components of time delay, finite differences, and signal summation) inspired by dynamical systems theory to replace the existing approach in time series models like PatchTST, Time-SSM, and ModernTCN, which segments time series channels into overlapping chunks. Adding these augmentations to the patching strategy, prior to feeding data into the transformer encoder, appears to enhance performance across tasks and reduce the need for parameter tuning.

**Strengths:**

The summary in Figure 3 is clear, easy to understand, and provides a strong overview of the work. The methods are clearly inspired by dynamical systems theory.

The experimental evaluation is reasonable, and appears to me consistent with norms in this area (particularly in comparison to Time-SSM and PatchTST), and addresses the research questions outlined in Section 5. The authors demonstrate that these alternative patching and embedding strategies can improve performance with smaller parameter counts, at least on the evaluated datasets. Additionally, the authors have provided code implementing their approach.

**Weaknesses:**

### Organization and Framing

The paper’s organization and framing are somewhat confusing. The main contribution seems to be an empirical evaluation of patching/embedding strategies (time delay, principal components of time delay, finite differences, and signal summation) that improve performance in models with smaller parameter counts. These strategies draw some mild inspiration from dynamical systems theory as a form of feature engineering to incorporate prior knowledge about sequences. This is a valid and valuable empirical contribution (discussed further later).

However, the introduction and theoretical framing are largely disconnected from the methodology in Section 4 and the experiments in Section 5. Problem formulation in Section 2 lacks alignment with the experimental evaluation, and the claim of establishing empirical and theoretical connections between dynamical systems and time series seems tenuous. The “embedding duality theory” (e.g., Section 4.2) is mathematically ambiguous, and the concepts introduced may be better suited to be introduced after strong empirical evidence. This is fine if the scope of the work is to just test and report on some ideas for how to improve the predictive performance using less parameters, but is unsubstantiated as a main claim of the paper.

I would suggest reframing the paper to start from the limitations of existing state-space and transformer models for time series (e.g., PatchTST) and why improvements in data augmentation or embedding are needed. The related work, especially Figure 3, already outlines some of this context. Following this, I guesses discuss how insights from dynamical systems inform these physics-guided embedding strategies, clarifying where they are effective or limited, and make the empirical case.

In Section 4.1, the formulation needs greater clarity. Please specify, mathematically, the exact input fed into the model as the embedding, particularly in the HD-Emb and ID-Emb sections, as this remains unclear. Including a section describing the architecture of existing state-space and transformer models and how your method builds on them may also help clarify.

The proposed strategies themselves are not novel; the use of integral and derivative signals for prediction has long been established in signal processing, dynamical systems, and time-series literature (for well over one hundred years). However, strong (positive or negative) empirical results could still offset this.

### Experimental Section:
RQ1 is insufficiently supported, as the embedding duality theory lacks precise definition. RQ2 and RQ3, however, are aligned with the experimental evaluations. Performance gains appear marginal overall (e.g., in Table 5) and inconsistent across datasets; for instance, electricity ETTh1 shows benefits with long sequences, while weather does not. While lines 378–490 offer a brief explanation, a stronger link between data properties and empirical results would be valuable (e.g., reporting statistics about physical properties the sequences and their ultimate relation to performance w.r.t. to embedding strategy), and would support the hypothesis put forth about physics guided embeddings. Additionally, though the physics-guided embeddings reportedly reduce parameter counts, this is not thoroughly analyzed. I think a more extensive evaluation than just the 8 datasets would be useful to prove this case. Overall, the connection of key relations between the known physical properties/statistics of each dataset, and what is empirically observed in terms of performance across different tasks/embeddings strategies is not clear in this version of the manuscript.

### Errata:
- Line 155: physics-guided.
- Line 161: Could you explain why $\tau$ should appear in the dimension?
- Line 171: multivariate

**Questions:**

- Could you give the mathematical formulation for each of the strategies in section 4.1? (i.e, the quantity you are passing to transformer), especially comparison to the formulation of patches/embedding used in the original Time-SSM, PatchTST and ModernTCN works.
- Line 298: what is meant by dynamical dimension?
- In Figure 5. what is on the y axis, what is the previous approach you are comparing to? Why is the resemblance between the datasets interesting? Could you elaborate on what it means to be significantly enhanced in on lines 291?
- In Figure 7. Could you provide more details/results about how this figure was produced? (what is the experimental context, and how you calculate the parameter counts of the physics guided embeddings, related to question Q1)?
- Could you summarize the key relations between the known physical properties/statistics of each dataset, and what you empirically observe across different tasks/embeddings strategies?

---

> ### Comment · Reviewer_LWUw · 2024-11-24
>
> Thanks to the authors for thorough response. Having reviewed the responses to my questions and those of the other reviewers and after careful consideration, I have decided to retain my score.
>
> My concern is the same is the other reviewer — is that the concept of embedding duality is still not well specified, so the posed research questions are not well supported. If the contents presented in Section 4.2 are a primary component of the work, it requires more development. Additionally, the clarity in general of the work is a concern to me.
>
> If the contribution is the claim that existing parameterized embedding layers in deep learning may not be necessary — which is a potentially valid claim— this must be made clear in the narrative of the paper (and the corresponding research question you ask). Then the paper should be clearly structured around empirically demonstrating this point, and providing the necessary detail.
>
> I maintain the opinion the manuscript needs substantial revision, as to ensure the claims are consistent with the mathematical formulation and the numerical experiments.

---

### Official Review · Reviewer_SFQb · 2024-11-03

**Soundness:** 4
**Presentation:** 1
**Contribution:** 3
**Rating:** 6
**Confidence:** 3

**Summary:**

The submission introduces embedding duality - parameterized embeddings are linear estimations of underlying non-linear dynamics. Thus, they claim that parameterized embeddings are unnecessary and introduce physics guided time series embeddings. These physics guided embeddings are simple transformations of existing data (shifting, stacking, derivatives etc.). The authors provide theoretical justification and extensive experiments with a few datasets comparing parameterized embeddings and physics-guided time series embeddings for classification, forecasting, imputation, and few/zero-shot learning.

They make several claims that are fairly well justified with theoretical and empirical results:
1. Parameterized embeddings are unnecessary and introduce physics guided time series embeddings. Through prior theoretical and empirical work, they make the case for this.
2. Physics-guided embeddings are more computationally efficient - they consistently maintain a good performance without meticulous hyper-parameter searching. The authors demonstrate this in Fig 7 by plotting performance versus speed.
3. Physics-guided embeddings have the capacity to encapsulate richer and more essential information, consequently amplifying the performance. Implied from empirical experiments.

**Strengths:**

The strengths of this paper lie in the radical idea that parameterized embeddings are unnecessary, and the author make a good case for this through extensive experiments. The authors also provide intuition to why the physics-guided embeddings perform better. Thus, the reader can infer how to design such embeddings.

**Weaknesses:**

The pre-dominant weaknesses of this submission lies in its writing and presentation. While the sentences were well written, the writing was not done in a manner where the key ideas, jargon, and notation were introduced before using them. In the theory section, there were no transitions between lemmas, propositions, etc.. It was difficult to follow and connect the dots. In the experiments section, there were significant details of experiments that were not outlined and clear (even in the appendix). Furthermore, there were also sections in the manuscript that were questionable about why they were included in the main text. Please see below for comments and suggestions. I would be willing to improve the score after addressing these writing issues.

**Questions:**

Here are some questions I have:
1. Clarification question: What is the usual size of parameterized embeddings and how does it compare to physics guided embeddings? based on my understanding, shifting and stacking time series data and their derivatives could yield very large embeddings.
2. Follow up to 1: forecasting w.r.t. input length: one central advantage of using physics-guided embeddings is it runs faster. you also mention that longer inputs lead to better improvements. could you elaborate on what the memory requirements for using longer and longer inputs are? does the speed eventually stop improving (and potentially degrade) when the inputs are extremely long? how does this impact the performance?
3. What was the rationale for including/introducing PC-Emb and ID-Emb in the main text when they were not being evaluated? If these results are not good, I would suggest putting them in the appendix/supplementary instead. Also, the page 4 discussion section is strangely placed and is required to justify exclusion of PC and ID methods. This should be rewritten somewhere else because there have been no results presented.
4. Why is the covariance matrix used for PC-Emb? It is not clear from the manuscript. Please elaborate on why this is the case.
5. Fig 4: why is parameterized embedding illustrated as PCA? Based on my understanding, the PCA embeddings are physics-guided embeddings as presented in section 4.
6. Section 5.1 (visualization): Please justify why this part is necessary. Based on my understanding of the embeddings, it is trivial to see that using physics-guided embeddings capture the dynamical structure better because they use simple transforms/stacking of existing data - and therefore there will be patterns in the data.
7. Section 5.1 (dim scaling law): Based on my understanding of the submission, the main point of the physics-guided embeddings is that it encapsulates intrinsic dynamical characteristics of the data. However, this part advocates for parameterized and could be irrelevant. Please justify why this part is relevant for the manuscript.
8. Section 5.1 (causal-drectional modeling): Please explain what this part aims to show. I’m confused why this is important.

Some comments related to formatting and writing:
- Page 2 (problem statement): What is dimension C? Please define.
- Page 2 (problem statement): A brief description of how the encoder, embeddings, and decoder fit together in the deep learning model would help clear up confusion. At first glance, because the authors introduced embeddings then encoder then decoder, the reader will be confused whether this is different to standard deep learning paradigms.
- Page 2 (problem statement): When introducing encoder, embedding, and decoder, it would be beneficial for the reader to describe what each component does. Also, what does “serves as a flexible architecture mean?”
- Page 2 (encoder): What does SSM mean? Please define these abbreviations.
- Figure 2: The figure is confusing because it is introduced before the different models are introudced. I would recommend either defining the methodsor placing this figure further down manuscript.
- Page 3 (bottom): Typo - physic-guided embeddings -> physics guided embeddings
- Section 4.2: There is lack of introduction and transitions between propositions, lemmas etc.. I would suggest describing the propositions/lemmas before stating them. There are also no transitions between the theory to conceptually connect them together. As a reader, I found it challenging to connect all the dots together.
- Fig 5: There are no details of how the left subplots for parameterized embeddings were generated. what does number of iterations mean? Please explain how these experiments were set up.
- Fig 5: The claim that physics guided embeddings have superior performance for downstream tasks is unwarranted because there have not been any results presented yet.
- Section 5.1 and 5.3: There are signifcant portions of the experimental setup missing from this section. I understand some of this information is the appendix, but the main body of the paper is not self-sufficient to understand the basic experimental setup. Please include some form of description of the experimental setups.
- Fig 8: the legends do not represent the colors and patterns of the bars. Please fix.
- Section 5.2 (forecasting w.r.t. various embedding techniques): These are good observations. I would have loved to see a comparison (possibly tabular) indicating when each embedding should be used.

---

### Official Review · Reviewer_uPZs · 2024-11-03

**Soundness:** 2
**Presentation:** 1
**Contribution:** 2
**Rating:** 3
**Confidence:** 2

**Summary:**

While the paper addresses an intriguing topic in time series modeling and proposes a novel approach, I found several aspects challenging to understand, especially regarding the theoretical foundation and the specific contributions of the proposed method. I am open to revising my review if further clarification is provided in response to my comments or if other reviewers offer insights that improve my understanding of the paper. Below, I outline my main concerns.

1. The Embedding Duality Theory is the central theoretical contribution of this paper. However, I find the explanation of this theory quite vague. The description in lines 54-56 introduces Embedding Duality, but the mathematical foundation is not clearly established. Although Section 4.2 provides some theoretical results, the connection between these results and the Embedding Duality concept remains unclear. I would recommend a more rigorous and detailed mathematical formulation of Embedding Duality to clarify its implications and relationship to the results in Section 4.2.

2. Regarding the proposed embedding techniques, I did not observe significant algorithmic innovation. Section 4.1 outlines several embedding methods (e.g., time delay embedding, principal component embedding, high-order derivatives), all of which are well-established in the field. Furthermore, there are important existing embedding techniques that are not mentioned in the paper, such as state space models and reservoir computing, which are also commonly used for capturing dynamics in time series data.

In conclusion, while the topic is interesting and relevant, the theoretical framework requires a clearer and more rigorous explanation, and the innovation in embedding techniques needs to be better established. I am open to revising my review if the authors can address these concerns and clarify the theoretical and algorithmic contributions.

**Strengths:**

This paper introduces the concept of "Embedding Duality Theory" in the context of time series analysis. The authors propose that a parameterized embedding layer in a deep learning model serves as a linear approximation of the underlying nonlinear dynamics in time series data. The paper suggests that, by using physics-based priors, the model can bypass traditional parameterized embeddings, resulting in reduced parameters, faster computation, and improved performance on various time series tasks. The methods are implemented as a plug-and-play module.

**Weaknesses:**

See the summary

**Questions:**

I would like to thank program chairs for the detailed suggestions to improve the review. Based on the guidance, here are my questions for the authors:

1. Formal Definition of Embedding Duality Theory:
The Embedding Duality Theory is presented as a key theoretical contribution of the paper, but it lacks a formal mathematical definition. Could the authors provide a formal mathematical definition of Embedding Duality? How does this theory relate to existing theories?

2. Connection Between Embedding Duality and Section 4.2 Results:
Section 4.2 contains theoretical results, yet the connection between these results and Embedding Duality Theory is unclear. Could the authors explicitly explain how the results in Section 4.2 support, derive from, or are otherwise linked to the Embedding Duality Theory? Any clarification on this alignment would help in understanding the theoretical contributions.

3. Innovation in Embedding Techniques:
In Section 4.1, the embedding methods discussed (such as time delay embedding, principal component embedding, and high-order derivatives) appear to be well-established. Could the authors clarify how the proposed physics-guided embeddings differ from or improve upon these traditional embedding methods? Are there any novel aspects in the application or combination of these techniques that enhance time series analysis? How does the approach compare to or differ from other modern embedding techniques, such as state space models and reservoir computing, which are known for capturing dynamics in time series data?

---

### Meta-Review · Area_Chair_M7TN · 2024-12-20

**Metareview:**

The paper introduces a new concept  "Embedding Duality Theory" in the context of time series analysis.
It aims to demonstrate that parameterized  embeddings in deep learning models  approximates the underlying nonlinear dynamics.
Based on the reviews, the rebuttal and the discussions, several missing points make the paper difficult to recommend for acceptance
at this points.

**Additional Comments On Reviewer Discussion:**

reviewers acknowledged rebuttals but think that concerns still exist about the paper.

---

### Decision · Program_Chairs · 2025-01-22

Reject